# ReTrace: Reinforcement Learning-Guided Reconstruction Attacks on Machine Unlearning

**Mengyao Ma**[1,2] **Shuofeng Liu**[1,2] **Minhui Xue**[2,3] **Surya Nepal**[2,3] **Guangdong Bai**[4]
[1]The University of Queensland  [2]CSIRO's Data61
[3]Responsible AI Research (RAIR) Centre, Adelaide University  [4]City University of Hong Kong
`{mengyao.ma,shuofeng.liu}@uq.edu.au`
`{jason.xue,surya.nepal}@data61.csiro.au`
`g.bai@cityu.edu.hk`

## Abstract

Machine unlearning has emerged as an inevitable AI mechanism to support GDPR requirements such as revoking user consent through the "right to be forgotten". However, existing approaches often leave residual traces that make them vulnerable to data reconstruction attacks. In this work, we propose ReTrace, the *first reconstruction attack framework* that uniquely formulates unlearned data recovery on *large-scale deep architectures* as a reinforcement learning (RL) problem. By treating residual unlearning *traces* as reward signals, ReTrace guides a generator to actively explore the input space and converge toward the forgotten data distribution. This RL-guided approach enables both *instance-level* recovery of individual samples and *distribution-level* reconstruction of unlearned classes. We provide a theoretical foundation showing that the RL objective converges to an exponential-tilted distribution that amplifies forgotten regions. Empirically, ReTrace achieves up to 73.1% instance-level recovery and reduces FID and KL scores beyond two state-of-the-art baselines. Strikingly, on the challenging task of text unlearning, it improves BLEU scores by nearly 100% over black-box baselines while preserving distributional fidelity, demonstrating that RL can recover even high-dimensional and structured modalities. Furthermore, ReTrace demonstrates effectiveness across both convolutional (ResNet) and transformer-based models, with Distil-BERT as the largest architecture attacked to date. These results show that current unlearning methods remain vulnerable, highlighting the need for robust and provably private mechanisms.

## 1 Introduction

Machine unlearning has recently emerged as a critical technique to empower users with the ability to remove their data from trained models, aligning with the *right to be forgotten* in GDPR (European Parliament & Council of the European Union, 2016) and responsible Artificial Intelligence (AI) innovation. To achieve this, early work on exact unlearning enforces deletion by retraining the model from scratch on the dataset with the target samples removed. While this approach provides a strong guarantee, it is computationally prohibitive for today's large and complex models (Ginart et al., 2019). As a result, recent research has shifted toward *approximate unlearning* (Bourtoule et al., 2021; Li et al., 2024), which directly modifies a trained model to erase the influence of unlearned data. These techniques offer more practical trade-offs between efficiency and unlearned strength, making machine unlearning a cornerstone of responsible AI innovation, where compliance with user rights and trustworthy data governance is essential (Liu et al., 2024).

Despite its promise, machine unlearning is vulnerable to *reconstruction attacks* (Zhang et al., 2023a; Bertran et al., 2024), where adversaries attempt to recover the data that was intended to be forgotten, as unlearning may have the unintended opposite effect: rather than concealing the data, it can inadvertently facilitate the localization of sensitive records in a vast corpus. As a result, it is easier for adversaries to search the needle in the haystack. Such attacks pose a direct threat to data privacy, as they might effectively reverse the unlearning process and expose sensitive information. The consequences can be severe in domains such as healthcare. For example, in a medical setting, even if a

patient requests deletion of their medical images or health records, a reconstruction attack could still recover identifiable details and compromise confidentiality. Such risk highlights the urgent need to systematically investigate the vulnerability of unlearning mechanisms under reconstruction attacks.

Several recent studies (Bertran et al., 2024; Pang et al., 2025; Hu et al., 2024) have explored reconstruction under unlearning. One line relies on *closed-form parameter analysis* (Bertran et al., 2024), which can exactly recover deleted samples but only applies to simple or linear models. Another line, *update-based reconstruction* (Hu et al., 2024), requires white-box gradients or parameter updates and is restricted to instance-level recovery, failing to generalize when multiple deletions occur. These limitations motivate the need for a framework that exploits unlearning traces in deep models, supports diverse access levels, and enables both instance- and distribution-level recovery.

**Our work**. In this work, we propose RETRACE, the first reconstruction attack framework that exposes privacy vulnerabilities in machine unlearning on both convolutional (ResNet (He et al., 2016)) and transformer-based (Distil-BERT (Sanh et al., 2019)) models, with Distil-BERT being the largest-scale architecture attacked to date. The key insight is that unlearning leaves detectable *traces* between pre- and post-unlearning models (Chen et al., 2025), which can be exploited with *reinforcement learning (RL)*. RETRACE operates in three steps. The first is *trace extraction*, which integrates signals such as output shifts, loss differences, and gradient alignments across model access levels, i.e., black-, grey-, and white-box. The second is *RL-guided reconstruction*, where a generator explores the input space and optimizes trace scores as rewards, enabling recovery even from complex models. The third is candidate selection and refinement, which supports both *instance-level* and *distribution-level* recovery.

We evaluate RETRACE through both theoretical analysis and empirical validation. On the theoretical side, we first establish that the RL objective converges to an exponential-tilted distribution, which provably amplifies regions with stronger unlearning traces. Building on this characterization, we show that RETRACE can successfully recover unlearned data at the instance level with high probability once a sufficient number of candidates are generated. Furthermore, using a bias–variance decomposition, we demonstrate that the empirical distribution of reconstructed samples converges toward the deleted-data distribution, with the bias controlled by the separability margin and the variance diminishing with the number of samples.

On the experimental side, we first visualize unlearning traces across different access levels and confirm that unlearned data leave consistent, instance-aligned residual signals. We then evaluate RETRACE on three benchmarking datasets under both exact and approximate unlearning. At the instance level, RETRACE can achieve the best success rate of 73.1% with an MSE of 0.17, demonstrating its ability to generate faithful reconstructions. At the distribution level, it can reduce the Fréchet Inception Distance (FID) to 99.1 and the Kullback-Leibler (KL) divergence to 2.53, showing strong alignment with the unlearned data distribution. Compared with baseline methods, RETRACE consistently attains lower MSE, higher success rates, and stronger feature similarity, confirming its superior effectiveness. Ablation study further demonstrates its robustness and generalization.

**Contributions**. Our main contributions are summarized as follows.

- **An impactful vulnerability unveiled on machine unlearning**. Most current unlearning techniques do not completely erase the influence of deleted data but instead leave discernible traces in the model. We find that these residual traces can be leveraged as reliable signals to reconstruct the supposedly unlearned data.

- **A novel reconstruction attack framework**. We propose RETRACE, a trace-guided RL-based reconstruction attack that systematically exploits unlearning-induced traces to recover unlearned data at both the instance and distribution levels. Different model access levels further make RETRACE more practical.

- **A comprehensive study and evaluation**. We provide a theoretical foundation for RETRACE in reconstructing the unlearned data at an instance level and distribution level, demonstrating its effectiveness. We also conduct comprehensive experimental evaluations to further show its performance.

## 2 PROBLEM STATEMENT

### 2.1 SYSTEM SETTING

**Unlearning settings**. We consider the standard machine unlearning setting. Let $\mathcal{D} = \mathcal{D}_{\text{ret}} \cup \mathcal{D}_{\text{del}}$ denote the training dataset, where $\mathcal{D}_{\text{ret}}$ represents the samples to be retained and $\mathcal{D}_{\text{del}}$ represents the samples requested to be deleted. A model is first trained on the full dataset $\mathcal{D}$, resulting in the *pre-unlearning model* $f^+$ with parameters $\theta^+$. Upon receiving a deletion request, an unlearning algorithm is applied to remove the influence of $\mathcal{D}_{\text{del}}$, producing the *post-unlearning model* $f^-$ with parameters $\theta^-$. The objective of unlearning is to ensure that $f^-$ behaves as if it were trained only on $\mathcal{D}_{\text{ret}}$, thereby eliminating any contribution of $\mathcal{D}_{\text{del}}$ to the model.

**RETRACE position**. We position RETRACE as a new form of reconstruction attack against machine unlearning. By leveraging RL, it progressively reconstructs the forgotten data (either at the instance level or distribution level) from residual traces left after unlearning.

### 2.2 THREAT MODEL

**Adversary knowledge**. We assume the adversary has access to both the pre-unlearning model $f^+$ and the post-unlearning model $f^-$, a scenario that arises in practice through model versioning, API updates, or common deployment practices (see Appendix A for evidence). Depending on the deployment, we consider three levels of model access. ***Black-box access:*** The adversary can query $f^+$ and $f^-$ and obtain their prediction outputs (e.g., labels and prediction probabilities). ***Grey-box access:*** In addition to outputs, the adversary can compute the loss with respect to the task loss function. ***White-box access:*** The adversary has full knowledge of model parameters $\theta^+$ and $\theta^-$, so that they can also obtain the gradient in training.

The adversary does *not* have access to the original training dataset $\mathcal{D}$ or the forgotten set $\mathcal{D}_{\text{del}}$. Instead, they may rely on an auxiliary public dataset $\mathcal{D}_{\text{pub}}$ drawn from similar distribution[1], which can be used to initialize candidate inputs for trace extraction.

**Attack goal**. The adversary's goal is to reconstruct the data belonging to the forgotten class. We consider two levels of reconstruction: (i) *instance-level*, where the adversary attempts to recover an individual sample in the unlearned class; and (ii) *distribution-level*, where the adversary aims to approximate the overall distribution of $\mathcal{D}_{\text{del}}$. Both types of recovery undermine the privacy guarantees of unlearning and expose sensitive information that should have been removed. Formally, the goal can be expressed as follows.

***Instance-level reconstruction***. Let $d_{\mathcal{X}} : \mathcal{X} \times \mathcal{X} \to \mathbb{R}_{\geq 0}$ be a task-appropriate metric. Given $\varepsilon > 0$, we say an adversary $\mathcal{A}$ succeeds at instance-level reconstruction if it outputs $\widehat{x} \in \mathcal{X}$ such that

$$\exists\, (x, y) \in \mathcal{D}_{\text{del}} \text{ with } d_{\mathcal{X}}(\widehat{x}, x) \leq \varepsilon. \tag{1}$$

For a $k$-set $\widehat{X} = \{\widehat{x}_1, \ldots, \widehat{x}_k\}$, success can be measured by the minimum matching distance (MMD) $\text{MMDist}(\widehat{X}, \mathcal{D}_{\text{del}}) = \min_\pi \frac{1}{k} \sum_{j=1}^{k} d_{\mathcal{X}}(\widehat{x}_j, x_{\pi(j)})$ and a threshold $\varepsilon$.

***Distribution-level reconstruction***. Let $\mathbb{P}_{\text{del}}$ denote the (unknown) distribution of inputs in the deleted set $\mathcal{D}_{\text{del}}$, and let $\widehat{\mathbb{P}}$ be the distribution induced by the reconstructed samples. We say the adversary achieves distribution-level reconstruction if the reconstructed distribution $\widehat{\mathbb{P}}$ is close to $\mathbb{P}_{\text{del}}$ under some statistical distance $d(\cdot, \cdot)$, i.e., $d(\widehat{\mathbb{P}}, \mathbb{P}_{\text{del}}) \leq \varepsilon$, for a small threshold $\varepsilon > 0$.

## 3 APPROACH

In this section, we delve into the internals of RETRACE. It is designed to reconstruct unlearned data by exploiting residual traces left after unlearning. Our insight is that when a model undergoes unlearning, the discrepancy between the pre-unlearning model $f^+$ and the post-unlearning model $f^-$ inevitably reveals subtle *unlearning traces* (Chen et al., 2025). These traces provide exploitable

---

[1]Similar distribution means the auxiliary public data can belong to the same semantic class as the forgotten sample, but not imply population-level access.

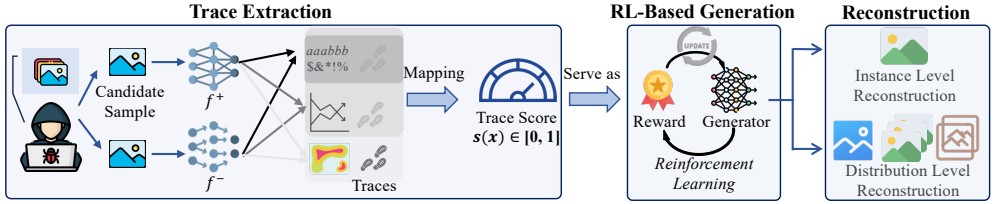

Figure 1: Overall workflow of RETRACE.

*signals* regarding the unlearned data. To leverage these signals effectively, we formulate the reconstruction problem as a *reinforcement learning (RL)* task. The learning agent explores the input space, and the feedback it receives, derived from unlearning traces, serves as the reward. By optimizing toward higher rewards, the agent gradually converges to regions of the input space that better approximate the unlearned data. The overall workflow of RETRACE is illustrated in Figure 1.

## 3.1 UNLEARNING TRACE EXTRACTION

The first step of RETRACE is to extract *unlearning traces*, which we define as *residual signals* that expose the behavioral differences between the pre-unlearning model $f^+$ and the post-unlearning model $f^-$. Since such differences are only manifested when the models are queried with inputs, we probe the models with a candidate input $x$ and extract the trace. Given $x$ drawn from a public distribution $\mathcal{D}_{\text{pub}}$ or sampled from the generator $\pi_\phi$, RETRACE quantifies traces at three progressively informative levels depending on the degree of model access. We follow (Sarmad et al., 2019) in designing trace extraction strategies tailored to different access levels.

**Prediction-level traces**. With black-box access, we measure the prediction discrepancy between $f^+$ and $f^-$ to check whether unlearning leaves residual influence on the output distribution. The trace is defined as the $\ell_2$ distance between their prediction vectors:

$$\delta_{\text{pred}}(x) = \|f^+(x) - f^-(x)\|_2. \tag{2}$$

**Loss-level traces**. With grey-box access, we extract traces from the task loss $\ell(\cdot)$. Using a pseudo-label $\hat{y}$ from $f^+$, we compute the absolute difference between the two models' losses:

$$\delta_{\text{loss}}(x) = \left|\ell(f^+(x), \hat{y}) - \ell(f^-(x), \hat{y})\right|. \tag{3}$$

**Gradient-level traces**. With white-box access, we capture differences in sensitivity to input perturbations. The trace is measured by the cosine distance between input gradients:

$$\delta_{\text{grad}}(x) = 1 - \cos\big(\nabla_x \ell(f^+(x), \hat{y}), \ \nabla_x \ell(f^-(x), \hat{y})\big). \tag{4}$$

Together, the unlearning trace of an input $x$ under different access levels can be represented as:

$$T(x) = \begin{cases} \big(\delta_{\text{pred}}(x)\big), & \text{black-box access,} \\ \big(\delta_{\text{pred}}(x), \ \delta_{\text{loss}}(x)\big), & \text{grey-box access,} \\ \big(\delta_{\text{pred}}(x), \ \delta_{\text{loss}}(x), \ \delta_{\text{grad}}(x)\big), & \text{white-box access.} \end{cases} \tag{5}$$

$T(x)$ is then used to guide the unlearned data reconstruction process.

## 3.2 RL-GUIDED GENERATION

**Formulation as RL**. RETRACE formulates data reconstruction as an RL problem in which a policy network actively explores the generator's latent space and is guided toward regions that preserve unlearning traces. The RL iteration begins by sampling a latent vector $z_0 \sim \mathcal{N}(0, I)$, representing an uninformed prior over the generator input space. A policy network $\pi_\theta$, (e.g., implemented as a multilayer perceptron (MLP)), transforms $z_0$ into a new latent vector $z_1 = \pi_\theta(z_0)$, which is then passed to the fixed generator $G$ to synthesize an image $x_1 = G(z_1)$. Through policy optimization with $\tau$ iterations, $x_\tau$ will be generated, representing the unlearned image.

The environment consists of the pre- and post-unlearning models $(f^+, f^-)$. For a generated sample $x_i$, the environment returns a scalar reward $s(x_i) = \texttt{min-max}(r(x_i))$, where

$$r(x_i) = \begin{cases} -\alpha\delta_{\text{pred}}(x_i), & \text{black-box,} \\ -\alpha\delta_{\text{pred}}(x_i) - \beta\delta_{\text{loss}}(x_i), & \text{grey-box,} \\ -\alpha\delta_{\text{pred}}(x_i) - \beta\delta_{\text{loss}}(x_i) - \gamma\delta_{\text{grad}}(x_i), & \text{white-box,} \end{cases} \qquad (6)$$

with $\alpha, \beta, \gamma \geq 0$ controlling the scale, and $s(x_i) \in [0, 1]$ is the normalized trace score. High trace scores indicate that $x_i$ lies closer to the forgotten data manifold, thus providing meaningful learning signals for the RL agent.

**Policy optimization via PPO**. To optimize the policy network $\pi_\theta$, we adopt an actor–critic version of Proximal Policy Optimization (PPO) (Schulman et al., 2017). Specifically, in the first training round, for synthesized sample $x_1 = G(z_1)$, its trace score $s(x_1)$ is first calculated, preparing for the following optimization, which starts from the second round. In subsequent rounds, PPO does not directly update the policy using the absolute reward $s(x_i)$; instead, it uses the *advantage*

$$A_{i+1} = s(x_i) - V_\psi(z_i), \qquad (7)$$

where $V_\psi$ is a critic network that estimates the expected trace score (reward) at latent state $z_i$. The critic is trained to regress toward $s(x_i)$ and therefore provides a learnable baseline that stabilizes the training dynamics by ensuring that updates depend on reward improvements. The sign of $A_{i+1}$ indicate whether the update direction is desirable. A positive advantage implies that the trace score obtained in the previous round exceeds the critic's expectation, meaning that the policy update has improved the generator's output and should continue in a similar direction. Conversely, a non-positive $A_{i+1}$ indicates that the optimization is suboptimal. When $|A_{i+1}|$ is small, the update yields little improvement, whereas a large negative advantage suggests that the update may move in the opposite direction of the desired behavior, so the policy should adjust its update direction accordingly.

Through each iteration, the clipped PPO surrogate objective is

$$\mathcal{L}_{\text{PPO}}(\theta) = \mathbb{E}_{z_i \sim \mathcal{N}(0,I)} \left[ \min\big(r_\theta(z_i) A_{i+1}, \ \text{clip}\big(r_\theta(z_i), 1 - \epsilon, 1 + \epsilon\big) A_{i+1}\big) \right], \qquad (8)$$

where $r_\theta(z_i)$ is the importance ratio between the new and old policies, and $\epsilon$ is a small positive clipping parameter (typically 0.1 or 0.2) that limits how much the new policy is permitted to diverge from the old one, ensuring stable and conservative updates. This objective encourages the actor to adjust $\pi_\theta$ such that the generated latent codes $z_{i+1} = \pi_\theta(z_i)$ yield larger trace scores than those predicted by the critic baseline. As training progresses, both $s(x_i)$ and the critic estimate $V_\psi(z_i)$ increase, and the advantage $A_{i+1}$ gradually approaches zero, indicating convergence. At convergence, the policy consistently steers latent vectors toward regions in which $G(z_{i+1})$ reconstructs samples that align closely with the unlearned data manifold.

By repeated interaction, the policy progressively reshapes the latent codes such that $G(z_\tau)$ yields images that maximize the discrepancy between $f^+$ and $f^-$. This guided exploration enables RE-TRACE to recover samples aligned with the unlearned data.

### 3.3 RECONSTRUCTION

After the RL stage, $\pi_\theta$ is well-trained. RETRACE generates $d_i{}_{i=1}^n$ reconstructed images by using $n$ independent initial latent vectors $\{z_i^{\text{in}}\}_{i=1}^n \sim \mathcal{N}(0, I)$. Then, for each $d_i$, their trace score $s(d_i)$ is calculated for further best reconstruction sample selection. Based on RETRACE's RL-guided generation, two levels of reconstruction can be carried out.

**Instance level**. At the instance level, we identify the top-scoring candidate

$$\hat{d} = \arg\max_{d_i} s(d_i), \qquad (9)$$

which serves as the best approximation of an unlearned instance.

**Distribution level**. At the distribution level, our goal is to approximate the forgotten data distribution. We first rank all generated candidates by their trace scores $\{s(d_i)\}_{i=1}^n$ and select the top-$k$ elements:

$$\widehat{\mathcal{D}}_{\text{forget}} = \{d_i \mid i \in I_k\}, \qquad (10)$$

where $I_k$ indexes the $k$ highest-scoring samples.

### 3.4 DISCUSSION ON EFFICIENCY

The overall complexity of RETRACE is $O(I \cdot N \cdot C_f + N \log N)$, dominated by generator sampling and trace evaluation across $I$ RL iterations with $N$ candidates per iteration. Here, $C_f$ denotes the cost of a single model forward/backward pass. In practice, the runtime is comparable to standard adversarial training or RL-based generation pipelines. Detailed explanation on efficiency is in Appendix B.

## 4 THEORETICAL ANALYSIS

In this section, we provide a theoretical foundation of RETRACE's attack effectiveness. Our goal is to formally establish that, under mild assumptions, RL optimization in RETRACE converges to policies that maximize unlearning traces, hence enabling effective reconstruction of the unlearned data distribution. Although RETRACE is implemented using PPO for stability, PPO's clipped surrogate objective does not correspond to a well-defined functional over policy distributions. Therefore, following standard practice in RL theory, we analyze a tractable KL-regularized surrogate objective that captures two essential behaviors of PPO—reward maximization and conservative policy updates—and is used solely as an analytical abstraction (see full proofs in Appendix C).

### 4.1 DEFINITIONS AND ASSUMPTIONS

**Definition 1** (Trace score). *Let $s : \mathcal{X} \to [0,1]$ denote the (fixed) trace score produced by the detector, quantifying the strength of unlearning traces for any $x \in \mathcal{X}$.*

**Definition 2** (Trace separability). *Let $\mathbb{P}_{\mathrm{del}}$ and $\mathbb{P}_{\mathrm{ret}}$ denote the input distributions of the deleted and retained data. Traces are separable if there exists a margin $\Delta > 0$ such that*

$$\mathbb{E}_{x \sim \mathbb{P}_{\mathrm{del}}}[s(x)] \geq \mathbb{E}_{x \sim \mathbb{P}_{\mathrm{ret}}}[s(x)] + \Delta. \tag{11}$$

**Assumption 1** (Policy expressiveness). *The policy class $\Pi$ contains densities absolutely continuous w.r.t. a public prior $p_0$ that can approximate $\mathbb{P}_{\mathrm{del}}$ in total variation to arbitrary precision. In particular, $\Pi$ includes the exponential-tilting family*

$$\pi_\lambda(x) \propto p_0(x) \exp(\lambda s(x)), \qquad \lambda \in \mathbb{R}. \tag{12}$$

**Assumption 2** (Abstract RL convergence). *The policy distribution $\pi \in \Pi$ is optimized through the RL procedure described in Section 3, and its evolution can be abstracted by the maximization of a strictly concave functional $\mathcal{J}_\tau(\pi)$ of the form*

$$\mathcal{J}_\tau(\pi) = \mathbb{E}_{x \sim \pi}[s(x)] - \tau \, \mathrm{KL}(\pi \,\|\, p_0), \qquad \tau > 0. \tag{13}$$

*We assume the optimization dynamics converge almost surely to a stationary point of $\mathcal{J}_\tau$ within $\Pi$.*

This functional is introduced solely as an analytical surrogate and is not required to coincide exactly with the implementation in Section 3; rather, it provides a tractable abstraction that captures the monotonic improvement in expected trace score and policy smoothness.

**Theorem 1** (Optimal policy form). *Under Assumptions 1 and 2, any stationary point $\pi^\star$ of the abstract functional $\mathcal{J}_\tau$ takes the form*

$$\pi^\star(x) \propto p_0(x) \exp(s(x)/\tau), \tag{14}$$

*and is the unique maximizer within the policy space $\Pi$.*

Theorem 1 proves the exponential-tilted form $\pi^\star \propto p_0 \, e^{s/\tau}$ by using a variational (Lagrangian) derivation under the normalization constraint together with the strict concavity of $\mathcal{J}_\tau$; the complete proof is provided in Appendix C.1.

**Theorem 2** (Instance-level reconstruction). *Let $\pi^\star$ be as in Theorem 1 and fix $\varepsilon > 0$. Define*

$$p_\star := \pi^\star\Big(\mathcal{N}_\varepsilon(\mathrm{supp}(\mathbb{P}_{\mathrm{del}}))\Big). \tag{15}$$

*Then for $k$ i.i.d. candidates drawn from $\pi^\star$ and refined by the local step in Section 3.3,*

$$\Pr\Big[\exists j \leq k : \min_{x \in \mathcal{D}_{\mathrm{del}}} d_{\mathcal{X}}(\hat{x}_j, x) \leq \varepsilon\Big] \geq 1 - (1 - p_\star)^k. \tag{16}$$

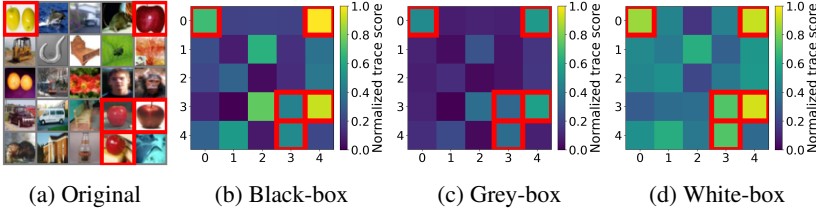

| (a) Original | (b) Black-box | (c) Grey-box | (d) White-box |

Figure 2: Traces of unlearned data under different model access levels on CIFAR-100 in the approximate unlearning scenario. The images in red boxes represent the unlearned data "*Apple*".

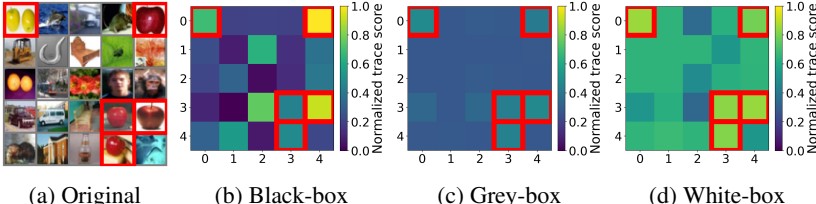

| (a) Original | (b) Black-box | (c) Grey-box | (d) White-box |

Figure 3: The normalized version of Figure 2 to show the advancement in Grey- and White-box.

Theorem 2 guarantees instance-level success by using the fact that $\pi^\star$ assigns positive probability mass to any $\varepsilon$-neighborhood of $\mathrm{supp}(\mathbb{P}_{\mathrm{del}})$ and a standard i.i.d. coverage bound $1 - (1 - p_\star)^k$; the full proof appears in Appendix C.2.

**Theorem 3** (Distribution-level reconstruction). *Let $\widehat{P}_k$ be the empirical distribution of the reconstructed candidates. Under Definitions 1, 2 and Assumptions 1, 2, there exists a bias term $C_1(\tau, \Delta)$ such that, for sufficiently large $k$,*

$$d(\widehat{P}_k, \mathbb{P}_{\mathrm{del}}) \leq C_1(\tau, \Delta) + \epsilon(k, \delta), \tag{17}$$

*with probability at least $1 - \delta$. Here $d(\cdot, \cdot)$ is a general statistical distance (e.g., Maximum Mean Discrepancy (MMD), Wasserstein), and $\epsilon(k, \delta)$ is the sampling error vanishing as $k \to \infty$.*

Theorem 3 establishes distribution-level reconstruction by using a bias–variance decomposition $d(\widehat{P}_k, \mathbb{P}_{\mathrm{del}}) \leq d(\widehat{P}_k, \pi^\star) + d(\pi^\star, \mathbb{P}_{\mathrm{del}})$, where the bias stems from exponential tilting (controlled by $(\tau, \Delta)$) and the sampling term decays with $k$; the detailed argument is given in Appendix C.3.

## 5 EVALUATION

### 5.1 EXPERIMENTAL SETUP FOR MAIN EXPERIMENTS

We evaluate RETRACE on three datasets: CIFAR-100 (Krizhevsky et al., 2009), Food-101 (Bossard et al., 2014), and PathMNIST (Yang et al., 2023), using ResNet-18 (He et al., 2016) classifiers with paired models $(f^+, f^-)$ obtained by both exact and single gradient (approximate) unlearning (Thudi et al., 2022). Unlearning is performed in a class-wise manner with class 0 as the forgotten category. A DCGAN-style (Radford et al., 2015) generator is used for image tasks, guided by an RL policy to exploit unlearning traces across black-, grey-, and white-box access levels. We assess reconstruction quality at the instance level using *mean squared error (MSE)*, *cosine similarity (CS)* [2], and *success rate (SR)*, and at the distribution level using *Fréchet Inception Distance (FID)* and *Kullback–Leibler (KL) divergence*. For comparison, we include two state-of-the-art baselines: Unlearning Inversion Attack (UIA) (Hu et al., 2024) and HRec (Bertran et al., 2024). Detailed experimental settings are in Appendix D.1.

### 5.2 TRACE VISUALIZATION

To empirically demonstrate the feasibility of using traces as reconstruction signals, we visualize the traces of unlearned data with heatmaps and further present their distributions, showing that the

---

[2] We extract features from the pool3 layer of an ImageNet-pretrained Inception-V3 network to compute cosine similarity.

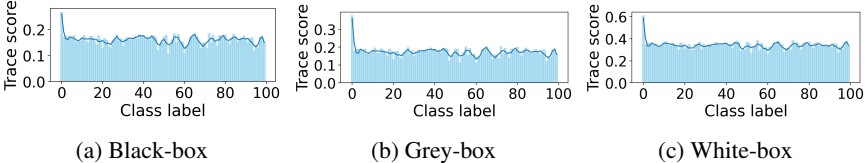

(a) Black-box        (b) Grey-box        (c) White-box

Figure 4: Trace distribution under different model access levels on CIFAR-100 in the exact unlearning scenario. Class 0 represents "*Apple*".

traces of unlearned data exhibit distinct patterns. Below, we explain the results on CIFAR-100 under approximate unlearning; the full evaluation results are provided in Appendix D.2.

**Trace of unlearned data**. Figure 2 visualizes unlearning traces on CIFAR-100 across three access settings. Panel (a) shows a $5 \times 5$ patch of original images with unlearned samples ("Apple") marked in red; Panels (b)–(d) present heatmaps of the normalized trace scores on the same grid. In the black-box setting, unlearned samples already exhibit top-quantile scores, showing that prediction-level discrepancies leave detectable signals. Grey-box access sharpens localization as loss-level differences enhance the contrast between forgotten and retained samples. White-box access further amplifies this effect: gradient-level information yields the clearest and most stable separation. Overall, unlearned data consistently leaves residual, instance-aligned traces across all access levels.

Additionally, to show the improvement in the white-box setting, Figure 3 provides a normalized version of Figure 2 that can make the contrast between the black-box, grey-box, and white-box settings more visually explicit. We have computed, for each $5 \times 5$ patch, the average trace value and then get the ratio between the highlighted red-square values and the corresponding patch average. We plot this visualization and show that the normalized ratios further amplify the relative contrast: the white-box and grey-box settings shows the largest ratio gap between unlearned and retained samples, while the black-box setting exhibits noticeably weaker ratios. This matches the quantitative behavior observed in our evaluation and reinforces the phenomenon that higher model access level contains more information regarding trace scores.

**Trace distribution**. We further analyze trace distributions across access levels. Figure 4 shows class-wise mean trace scores on CIFAR-100, with class 0 as the unlearned category "Apple". In the black-box setting, class 0 already scores higher than others, indicating measurable separation from prediction-level discrepancies. Grey-box access enlarges this gap by incorporating loss-level information, while white-box access further sharpens the contrast using gradient-level traces. Overall, the unlearned class consistently exhibits distinctly higher trace values, confirming that residual traces extend beyond instances to distributional patterns.

### 5.3 EFFECTIVENESS OF RETRACE

We first benchmark the instance-level and distribution-level reconstruction capabilities of RETRACE under three model access settings across three benchmarking datasets, and then compare its performance with two state-of-the-art reconstruction attacks. Figure 5 provides some generated cases by RETRACE, and more cases are presented in Appendix D.3.

(a) Instance.      (b) Distribution.

Figure 5: Generated unlearned samples ("Apple class") in CIFAR-100 under exact unlearning.

**Benchmarking**. For instance-level recovery, Table 1 shows that reconstructed samples achieve MSE values close to the intra-class criteria (e.g., CIFAR-100: 0.20 vs. 0.16), with SR around 50–60% under exact unlearning and further increasing under approximate unlearning (e.g., CIFAR-100: 73.1%). Meanwhile, CS values remain close to the criteria (e.g., Food-101: 0.49 vs. 0.54), indicating that recovered images preserve semantic similarity to the original unlearned data. These results confirm that RETRACE can reliably regenerate unlearned samples at the instance level. Since the results presented in Table 1 reflect the mean of the 5 runs, Table 6 in Appendix D.5 further shows their standard deviation.

At the distribution level, Figure 6 reports FID and KL divergence. Even under restrictive black-box access, RETRACE achieves reasonable alignment with the deleted distribution (e.g., CIFAR-

Table 1: The effectiveness of RETRACE from instance-level reconstruction

| Dataset | Exact Unlearning | | | | | | | | | Approximate Unlearning | | | | | | | | |
| | Black-box | | | Grey-box | | | White-box | | | Black-box | | | Grey-box | | | White-box | | |
| | MSE | SR | CS | MSE | SR | CS | MSE | SR | CS | MSE | SR | CS | MSE | SR | CS | MSE | SR | CS |
|---|---|---|---|---|---|---|---|---|---|---|---|---|---|---|---|---|---|---|
| CIFAR-100 | 0.23 | 52.3% | 0.43 | 0.22 | 58.7% | 0.46 | 0.20 | 62.4% | 0.47 | 0.24 | 55.9% | 0.46 | 0.21 | 68.7% | 0.49 | 0.17 | 73.1% | 0.50 |
| Food-101 | 0.25 | 48.9% | 0.38 | 0.23 | 54.7% | 0.42 | 0.23 | 50.4% | 0.44 | 0.20 | 56.6% | 0.41 | 0.18 | 62.5% | 0.49 | 0.16 | 65.3% | 0.49 |
| PathMNIST | 0.26 | 49.8% | 0.26 | 0.23 | 50.6% | 0.27 | 0.22 | 52.8% | 0.31 | 0.24 | 51.9% | 0.28 | 0.20 | 54.6% | 0.28 | 0.19 | 59.7% | 0.33 |

Criteria: CIFAR-100 (MSE = 0.16, CS = 0.57); Food-101 (MSE = 0.13, CS = 0.54); PathMNIST (MSE = 0.13, CS = 0.41).

Table 2: Effectiveness comparison of RETRACE and baselines under white-box model access.

| Dataset | Exact Unlearning | | | | | | | | | Approximate Unlearning | | | | | | | | |
| | UIA | | | HRec | | | RETRACE (Ours) | | | UIA | | | HRec | | | RETRACE (Ours) | | |
| | MSE | SR | CS | MSE | SR | CS | MSE | SR | CS | MSE | SR | CS | MSE | SR | CS | MSE | SR | CS |
|---|---|---|---|---|---|---|---|---|---|---|---|---|---|---|---|---|---|---|
| CIFAR-100 | 0.35 | 46.7% | 0.42 | 0.27 | 41.9% | 0.31 | 0.20 | 62.4% | 0.47 | 0.33 | 59.5% | 0.39 | 0.32 | 43.0% | 0.33 | 0.17 | 73.1% | 0.50 |
| Food-101 | 0.33 | 36.2% | 0.38 | 0.46 | 27.4% | 0.31 | 0.23 | 50.4% | 0.44 | 0.31 | 41.4% | 0.40 | 0.45 | 30.2% | 0.33 | 0.16 | 65.3% | 0.49 |
| PathMNIST | 0.43 | 41.0% | 0.26 | 0.42 | 33.6% | 0.22 | 0.22 | 52.8% | 0.31 | 0.39 | 37.4% | 0.24 | 0.44 | 37.1% | 0.20 | 0.19 | 59.7% | 0.33 |

100: FID = 142.1, KL = 3.62). Adding loss-level traces in the grey-box setting further reduces both FID and KL, and the white-box setting yields the best reconstruction quality (e.g., CIFAR-100: FID = 108.9, KL = 2.49; PathMNIST: FID = 96.5, KL = 2.22). Approximate unlearning consistently improves results, with lower FID and KL across datasets (e.g., CIFAR-100: FID = 99.1; PathMNIST: KL = 2.00), highlighting that residual traces left by approximate methods are easier to exploit. Together, these findings demonstrate that RETRACE achieves robust recovery of unlearned data across both granular instance-level signals and broader distributional structures. The detailed analyses are presented in Appendix D.3.

**Comparison with baseline methods**. Table 2 compares RETRACE with UIA (Hu et al., 2024) and HRec (Bertran et al., 2024) under the white-box setting, as both baselines are restricted to this access level. In the exact unlearning scenario, RETRACE achieves lower MSE (e.g., CIFAR-100: 0.20 vs. 0.35 for UIA and 0.27 for HRec), higher SR (e.g., PathMNIST: 52.8% vs. 41.0% for UIA), and higher CS (e.g., Food-101: 0.44 vs. 0.38 for UIA and 0.31 for HRec), indicating more faithful pixel- and feature-level recovery. Under approximate unlearning, RETRACE further outperforms both baselines across all metrics, with lower MSE (e.g., PathMNIST: 0.19 vs. 0.39 for UIA and 0.44 for HRec), higher SR (e.g., CIFAR-100: 73.1% vs. 59.5% and 43.0%), and the highest CS (e.g., Food-101: 0.49 vs. 0.40 and 0.33). Overall, RETRACE consistently surpasses UIA and HRec, demonstrating superior effectiveness in reconstructing unlearned data.

We also conduct experiments to compare our methods with baselines on distribution-level reconstruction. The results show that RETRACE still outperforms UIA and HRec in terms of KL and FID, demonstrating its effectiveness. The detailed results are listed in Table 6 in Appendix D.4.

## 5.4 ABLATION STUDY

We conduct ablation studies to demonstrate the robustness and generalization of RETRACE. We also discuss its potential limitations in Appendix D.11.

**Robustness**. Table 3 presents the ablation results on three unlearned classes from CIFAR-100 (*Aquarium fish*, *Bed*, and *Bridge*) under the white-box setting. Across all three categories, RETRACE delivers strong performance on all metrics: MSE remains low (0.21–0.23 for exact unlearning and 0.16–0.19 for approximate unlearning), SR consistently stays above 59%

Table 3: The effectiveness of RETRACE on different unlearned class reconstruction.

| Class ID | Exact Unlearning | | | Approximate Unlearning | | |
| | MSE | SR | CS | MSE | SR | CS |
|---|---|---|---|---|---|---|
| Aquarium fish | 0.21 | 59.1% | 0.41 | 0.18 | 70.2% | 0.44 |
| Bed | 0.22 | 60.5% | 0.42 | 0.19 | 69.2% | 0.43 |
| Bridge | 0.23 | 60.8% | 0.42 | 0.16 | 71.6% | 0.42 |

and rises to around 70% in the approximate setting, and CS values remain stable in the range of 0.41–0.44. These results demonstrate that RETRACE maintains robust reconstruction ability across different unlearned data classes. Some generated cases are listed in Figure 18 in Appendix D.7.

Figure 6: FID and KL scores of RETRACE from distribution-level reconstruction

**Generalization**. For text reconstruction (Ma et al., 2023b), we employ a GPT-2-based (Radford et al., 2019) generator, and a DistilBERT classifier (Sanh et al., 2019) serves as the unlearned model under the exact unlearning setting. We evaluate performance on the AG News (Zhang et al., 2015) dataset using two metrics: BLEU (Papineni et al., 2002), which mea-

Table 4: The effectiveness of RETRACE on text reconstruction

| Unlearned Class | Black-box | | Gray-box | | White-box | |
|---|---|---|---|---|---|---|
| | BLEU | MMD | BLEU | MMD | BLEU | MMD |
| Sports | 2.8 | 0.40 | 3.6 | 0.36 | 4.1 | 0.31 |
| World | 2.2 | 0.43 | 3.9 | 0.42 | 4.3 | 0.37 |
| Business | 2.6 | 0.46 | 3.1 | 0.46 | 4.6 | 0.32 |
| Sci/Tech | 2.0 | 0.41 | 3.7 | 0.43 | 4.0 | 0.39 |

sures content similarity between generated and target texts, and MMD (Gretton et al., 2012), which assesses the distributional alignment across classes. In Table 4, RETRACE consistently improves with stronger access levels. For example, BLEU increases from 2.8 (Sports, black-box) to 4.6 (Business, white-box), while MMD decreases across all classes (e.g., 0.46 to 0.32 for Business), indicating progressively better distributional fidelity.

# 6 RELATED WORK

**Machine unlearning**. Machine unlearning (Guo et al., 2020; Bourtoule et al., 2021; Chen et al., 2022; Thudi et al., 2022; Liu et al., 2025b) enables models to forget specific training data. Existing methods are either exact or approximate: exact unlearning retrains on the retained dataset with strong guarantees but high cost (Ginart et al., 2019; Guo et al., 2020), while approximate unlearning updates parameters or gradients for efficiency (Bourtoule et al., 2021; Jia et al., 2024a; Li et al., 2024). Techniques include partition-based retraining (Bourtoule et al., 2021), gradient adjustment (Thudi et al., 2022), and adaptive methods (Gupta et al., 2021), with extensions to graphs and LLMs (Chen et al., 2022; Jia et al., 2024b; Liu et al., 2025b). Machine unlearning is often evaluated using membership inference attacks (Shokri et al., 2017; Ma et al., 2023a; Zhang et al., 2023b) to measure residual memorization.

**Reconstruction attacks**. Reconstruction attacks aim to recover sensitive training data, first studied via model inversion and gradient leakage (Fredrikson et al., 2015; Zhang et al., 2023a; Bertran et al., 2024; Pang et al., 2025; Ma et al., 2024; Feng et al., 2024). In unlearning, this risk is amplified, as pre–post model differences can intensify leakage (Liu et al., 2025c). Existing studies either analyze parameter shifts, which succeed in linear models but fail on deep networks, or exploit updates and gradients to optimize synthetic inputs (Ginart et al., 2019; Bourtoule et al., 2021). While recent inversion attacks show that model differentials can expose unlearned data, current approaches remain limited to instance-level recovery and do not scale to distribution-level scenarios.

# 7 CONCLUSION

We propose RETRACE, a reconstruction attack framework that systematically exploits residual traces left by machine unlearning. By identifying traces across access levels and leveraging them as rewards, RETRACE enables both instance- and distribution-level reconstruction. To our knowledge, this is the first demonstration of reconstruction attacks on large-scale architectures, showing effectiveness on ResNet-18 for vision and Distil-BERT for text. We provide theoretical guarantees of convergence to an exponential-tilted policy that amplifies high-trace regions, ensuring reliable recovery, and validate the approach with extensive experiments across datasets, unlearned classes, models, and unlearning methods. Our findings expose critical vulnerabilities of current unlearning techniques and highlight the need for robust unlearning mechanisms, positioning RETRACE as both a principled framework for analyzing privacy risks and a practical benchmark.

ACKNOWLEDGEMENTS

We thank the anonymous reviewers for their insightful comments to improve this manuscript. Jason Xue and Surya Nepal are supported in part by the Responsible AI Research (RAIR) Centre, a landmark collaboration between Australia's national science agency, CSIRO, and Adelaide University, in partnership with the South Australian Government. This work was in part done when Guangdong Bai was with University of Queensland.

ACKNOWLEDGEMENT OF LLM USE

The use of the LLM in this work was strictly confined to linguistic refinement, including grammar and readability. All intellectual and technical contributions are the sole responsibility of the authors.

ETHICS STATEMENT

Our research adheres to the ethical principles outlined by the ICLR Code of Ethics. All experiments in this work are conducted on publicly available and fully anonymized datasets, including CIFAR-100, Food-101, PathMNIST, and AG News. These datasets are widely used in the machine learning community and do not contain personally identifiable information (PII) or sensitive attributes directly linked to real individuals. For text-based tasks, the corpus consists of news articles already curated for research purposes, where all identifying information has been removed. For medical data (i.e., PathMNIST), the dataset is part of the MedMNIST collection, which is designed to be de-identified and ethically suitable for public use. Therefore, this research does not involve the collection, processing, or exposure of human-subject data, and Institutional Review Board (IRB) approval is not required.

We further emphasize that our work focuses on methodological contributions to the study of machine unlearning and reconstruction attacks. The purpose of this research is to analyze vulnerabilities under controlled and responsible experimental settings, and to call for defenses and mitigations in subsequent studies. All experimental results are reported for the sole purpose of advancing scientific understanding of unlearning privacy issues and informing the development of more trustworthy AI systems.

REPRODUCIBILITY STATEMENT

We have taken several steps to ensure the reproducibility of our work. All datasets used in this study are publicly available benchmarks, including CIFAR-100, Food-101, PathMNIST, and AG News. The links for dataset access are provided in our paper. For theoretical analysis, all assumptions, definitions, and theorems are listed in Section 4. The complete proofs are provided in the Appendix C. The experimental settings, including unlearning techniques, model architectures, RE-TRACE parameter configurations, and training schedules, are fully detailed in Sections 5.1, D.1, and D.10. We report results across multiple datasets and model-access levels (black-, grey-, and white-box), and also provide ablation studies and text generalization experiments to demonstrate robustness. An repository with our method implementation and training scripts is available at: https://github.com/Trusted-System-Lab/ReTrace.

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

## A  THREAT MODEL

**Adversary knowledge.** We assume the adversary has access to both the pre-unlearning model $f^+$ and the post-unlearning model $f^-$. While this may appear strong, it is realistic in practice.

*API versioning in production.* Providers such as OpenAI expose multiple model versions simultaneously (e.g., `gpt-4-1106-preview` vs. `gpt-4-0125-preview`), where clients can query both before and after updates (OpenAI). Similarly, Google Cloud Vision and Microsoft Azure Cognitive Services maintain legacy API versions for backward compatibility (Google Cloud, b; Microsoft Azure). These setups allow adversaries to monitor output differences across versions, directly corresponding to access to $(f^+, f^-)$.

*Deployment practices.* Large-scale platforms such as Meta and Google routinely use shadow testing and canary rollouts, where both old and new models serve traffic concurrently for days (Meta Engineering; Google Cloud, a). In regulated industries (e.g., finance, healthcare), rollback readiness is mandatory, and older checkpoints are preserved and potentially exposed through misconfiguration or insider threats (NIST, 2023).

Together, these practices show that simultaneous access to $(f^+, f^-)$ is not only plausible but frequently realized in real-world AI deployments (Liu et al., 2025a).

## B  RETRACE'S EFFICIENCY

We mathematically analyze the computational overhead of RETRACE. Its computational complexity arises from three stages. First, *trace extraction* requires evaluating discrepancies between the pre-

Table 5: Average runtime per optimization component.

| Step | Average Time |
|------|-------------|
| PPO training (per optimization step) | 0.02 s |
| DCGAN each generation | 0.005 s |

and post-unlearning models. For each input $x$, computing the prediction-, loss-, and gradient-level traces involves forward and at most one backward propagation through the model, which incurs $O(C_f)$ cost, where $C_f$ denotes the per-sample model evaluation cost. Second, in the *RL-based generation* stage, each policy update requires drawing $N$ candidate samples from the generator and querying the detector for their trace scores, leading to a per-iteration complexity of $O(N \cdot C_f)$. With $I$ RL iterations, this stage totals $O(I \cdot N \cdot C_f)$. Finally, the *reconstruction* step involves ranking and refining the $N$ candidates, which can be done in $O(N \log N)$ time.

Overall, the end-to-end complexity of RETRACE is

$$O(I \cdot N \cdot C_f + N \log N), \tag{18}$$

dominated by the repeated generator sampling and trace evaluation. In practice, since $C_f$ corresponds to a single model forward/backward pass, the runtime remains comparable to standard adversarial training or reinforcement-learning-based generation pipelines. Table 5 provides the results of runtime for RETRACE per round.

## C   DETAILED PROOFS

### C.1   PROOF OF THEOREM 1

*Proof.* We analyze the abstract optimization problem, which serves as the theoretical surrogate introduced in Assumption 2.

$$\max_{\pi \in \Pi} \left\{ \mathcal{J}_\tau(\pi) = \int s(x)\pi(x)\,dx - \tau \int \pi(x) \log \frac{\pi(x)}{p_0(x)}\,dx \right\} \quad \text{s.t.} \quad \int \pi(x)\,dx = 1,\ \pi \geq 0. \tag{19}$$

Introduce a Lagrange multiplier $\lambda$ and define

$$\mathcal{L}(\pi, \lambda) = \int \left[ s(x)\pi(x) - \tau\,\pi(x) \log \frac{\pi(x)}{p_0(x)} \right] dx - \lambda \left( \int \pi(x)\,dx - 1 \right). \tag{20}$$

For any admissible direction $h$ with $\int h = 0$, the Gâteaux derivative is

$$\frac{d}{d\epsilon} \mathcal{L}(\pi + \epsilon h, \lambda) \Big|_{\epsilon=0} = \int \left[ s(x) - \tau \left( \log \frac{\pi(x)}{p_0(x)} + 1 \right) - \lambda \right] h(x)\,dx. \tag{21}$$

Stationarity for all $h$ yields the Euler–Lagrange condition

$$s(x) - \tau \left( \log \frac{\pi^\star(x)}{p_0(x)} + 1 \right) - \lambda = 0 \implies \log \frac{\pi^\star(x)}{p_0(x)} = \frac{s(x)}{\tau} - \frac{\lambda}{\tau} - 1, \tag{22}$$

hence

$$\pi^\star(x) = C\,p_0(x) \exp\big(s(x)/\tau\big), \qquad C = \exp\big(-\lambda/\tau - 1\big) = \left( \int p_0(x)\,e^{s(x)/\tau}\,dx \right)^{-1}. \tag{23}$$

**Strict concavity and uniqueness.** Consider the feasible set $\mathcal{P} = \{\pi \geq 0 : \int \pi = 1\}$, which is convex. The functional $\pi \mapsto \int s\,\pi$ is linear, and the negative KL term is strictly concave:

$$\forall \pi_1, \pi_2 \in \mathcal{P},\ \forall \alpha \in (0, 1): \quad -\tau \int \big[ \alpha\pi_1 + (1 - \alpha)\pi_2 \big] \log \frac{\alpha\pi_1 + (1 - \alpha)\pi_2}{p_0}\,dx$$
$$> -\tau \left[ \alpha \int \pi_1 \log \tfrac{\pi_1}{p_0}\,dx + (1 - \alpha) \int \pi_2 \log \tfrac{\pi_2}{p_0}\,dx \right]. \tag{24}$$

where the strict Jensen inequality for $\xi \log \xi$ applies unless $\pi_1 = \pi_2$ a.e. Therefore $\mathcal{J}_\tau$ is strictly concave over $\mathcal{P}$, implying the stationary solution $\pi^\star$ is the unique global maximizer in $\Pi$. This completes the proof. □

## C.2 PROOF OF THEOREM 2

*Proof.* Fix $\varepsilon > 0$ and define the $\varepsilon$-neighborhood of the deleted support

$$A_\varepsilon := \mathcal{N}_\varepsilon\big(\mathrm{supp}(\mathbb{P}_{\mathrm{del}})\big) = \big\{x \in \mathcal{X} : \exists x_0 \in \mathrm{supp}(\mathbb{P}_{\mathrm{del}}) \text{ s.t. } d_\mathcal{X}(x, x_0) \le \varepsilon\big\}. \tag{25}$$

Because $\mathbb{P}_{\mathrm{del}}(A_\varepsilon) > 0$ for any $\varepsilon > 0$ and $\mathbb{P}_{\mathrm{del}} \ll p_0$ by Assumption 1, we have $p_0(A_\varepsilon) > 0$. From Theorem 1, $\pi^\star(x) = Z_\tau^{-1} p_0(x) \exp(s(x)/\tau)$ with $Z_\tau = \int p_0 e^{s/\tau}$, hence

$$p_\star := \pi^\star(A_\varepsilon) = \frac{\int_{A_\varepsilon} p_0(x) e^{s(x)/\tau} \, dx}{\int_\mathcal{X} p_0(x) e^{s(x)/\tau} \, dx} \ge \frac{e^{\inf_{x \in A_\varepsilon} s(x)/\tau} p_0(A_\varepsilon)}{e^{\sup_{x \in \mathcal{X}} s(x)/\tau}} > 0, \tag{26}$$

since $s \in [0, 1]$ and $p_0(A_\varepsilon) > 0$. Let $X_1, \ldots, X_k$ be i.i.d. draws from $\pi^\star$. The event $\{X_j \notin A_\varepsilon, \ \forall j\}$ has probability $(1 - p_\star)^k$. Therefore

$$\Pr\Big[\exists j \le k : X_j \in A_\varepsilon\Big] = 1 - (1 - p_\star)^k. \tag{27}$$

By definition of $A_\varepsilon$, for any such $X_j$ there exists $x \in \mathcal{D}_{\mathrm{del}}$ with $d_\mathcal{X}(X_j, x) \le \varepsilon$. Let the refinement operator be denoted $\mathcal{R}$; assume it is $L$-Lipschitz and uses stepsize $\eta > 0$ that respects the neighborhood, i.e.,

$$\|\mathcal{R}(X_j) - X_j\|_\mathcal{X} \le \eta \quad \text{and} \quad \eta \le \varepsilon - d_\mathcal{X}(X_j, \mathcal{D}_{\mathrm{del}}). \tag{28}$$

Then $\hat{X}_j := \mathcal{R}(X_j) \in A_\varepsilon$ and $\min_{x \in \mathcal{D}_{\mathrm{del}}} d_\mathcal{X}(\hat{X}_j, x) \le \varepsilon$. Hence

$$\Pr\Big[\exists j \le k : \min_{x \in \mathcal{D}_{\mathrm{del}}} d_\mathcal{X}(\hat{X}_j, x) \le \varepsilon\Big] \ge 1 - (1 - p_\star)^k. \tag{29}$$

Finally, to attain target confidence $1 - \delta$, it suffices to choose $k \ge \log(1/\delta)/p_\star$. This proves the claim. $\qquad\square$

## C.3 PROOF OF THEOREM 3

*Proof.* Let $\pi^\star$ be the unique maximizer given by Theorem 1. Let $\widehat{P}_k := \frac{1}{k} \sum_{j=1}^k \delta_{\hat{X}_j}$ be the empirical distribution of the $k$ reconstructed samples after the local refinement $\mathcal{R}$.

**Step 1: Bias–variance decomposition for a general IPM.** Let $d(\cdot, \cdot)$ be any integral probability metric (IPM) induced by a function class $\mathcal{F}$:

$$d(P, Q) := \sup_{f \in \mathcal{F}} \Big| \mathbb{E}_P[f] - \mathbb{E}_Q[f] \Big|. \tag{30}$$

By the triangle inequality,

$$d(\widehat{P}_k, \mathbb{P}_{\mathrm{del}}) \le d(\widehat{P}_k, \pi^\star) + d(\pi^\star, \mathbb{P}_{\mathrm{del}}). \tag{31}$$

Moreover, if the refinement $\mathcal{R}$ is $L$-Lipschitz and maps pre-selected candidates $\widetilde{X}_j$ to $\hat{X}_j = \mathcal{R}(\widetilde{X}_j)$, then standard stability for IPMs on bounded domains implies (for a Lipchitz-bounded function class $\mathcal{F}$)

$$d(\widehat{P}_k, \pi^\star) \le C_L \, d(\widetilde{P}_k, \pi^\star), \qquad \widetilde{P}_k := \frac{1}{k} \sum_{j=1}^k \delta_{\widetilde{X}_j}, \tag{32}$$

for some constant $C_L \ge 1$ depending only on $L$ and the diameter of $\mathcal{X}$. Thus it suffices to control $d(\widetilde{P}_k, \pi^\star)$ and $d(\pi^\star, \mathbb{P}_{\mathrm{del}})$.

**Step 2: Sampling (variance) term** $d(\widetilde{P}_k, \pi^\star)$. For common choices of $d$, we have non-asymptotic concentration with explicit rates:

- **MMD.** If $\mathcal{F}$ is the unit ball of an RKHS with bounded kernel $k$ and $\sup_x k(x, x) \le K$, then by standard RKHS concentration (e.g., via McDiarmid), for any $\delta \in (0, 1)$,

$$\Pr\left[ \mathrm{MMD}(\widetilde{P}_k, \pi^\star) \le 2\sqrt{\frac{K \log(2/\delta)}{k}} \right] \ge 1 - \delta. \tag{33}$$

- $W_1$. If $\mathcal{X} \subset \mathbb{R}^d$ is bounded with diameter $D$, the Fournier–Guillin bound gives constants $C_d > 0$ such that, for any $\delta \in (0, 1)$,

$$\Pr\left[ W_1(\widetilde{P}_k, \pi^\star) \leq C_d \left( \frac{\log(2/\delta)}{k} \right)^{1/d} \right] \geq 1 - \delta. \tag{34}$$

For a general IPM with functions bounded by $\|f\|_\infty \leq B$, symmetrization plus Hoeffding inequality yields the generic bound

$$\Pr\left[ d(\widetilde{P}_k, \pi^\star) \leq 2\mathfrak{R}_k(\mathcal{F}) + B\sqrt{\frac{2\log(2/\delta)}{k}} \right] \geq 1 - \delta, \tag{35}$$

where $\mathfrak{R}_k(\mathcal{F})$ is the empirical Rademacher complexity (often $O(1/\sqrt{k})$).

**Step 3: Bias term $d(\pi^\star, \mathbb{P}_{\mathrm{del}})$ via exponential tilting.** From Theorem 1, $\pi^\star$ has density

$$\pi^\star(x) = \frac{p_0(x)\exp(s(x)/\tau)}{Z_\tau}, \qquad Z_\tau := \int p_0(x)\exp(s(x)/\tau)\,dx. \tag{36}$$

We first derive an upper bound for $d_{\mathrm{TV}}(\pi^\star, \mathbb{P}_{\mathrm{del}})$ via an upper bound on $\mathrm{KL}(\mathbb{P}_{\mathrm{del}}\|\pi^\star)$ and then apply Pinsker. By the chain rule for KL,

$$\mathrm{KL}(\mathbb{P}_{\mathrm{del}}\|\pi^\star) = \mathrm{KL}(\mathbb{P}_{\mathrm{del}}\|p_0) - \frac{1}{\tau}\mathbb{E}_{\mathbb{P}_{\mathrm{del}}}[s] + \log Z_\tau. \tag{37}$$

Since $s \in [0, 1]$, Hoeffding's lemma gives (for any base measure)

$$\log Z_\tau = \log \mathbb{E}_{p_0}\left[\exp\left(\tfrac{s}{\tau}\right)\right] \leq \frac{\mathbb{E}_{p_0}[s]}{\tau} + \frac{1}{8\tau^2}. \tag{38}$$

Plugging equation 38 into equation 37 yields the explicit upper bound

$$\mathrm{KL}(\mathbb{P}_{\mathrm{del}}\|\pi^\star) \leq \mathrm{KL}(\mathbb{P}_{\mathrm{del}}\|p_0) + \frac{1}{\tau}\left(\mathbb{E}_{p_0}[s] - \mathbb{E}_{\mathbb{P}_{\mathrm{del}}}[s]\right) + \frac{1}{8\tau^2}. \tag{39}$$

By Pinsker's inequality,

$$d_{\mathrm{TV}}(\pi^\star, \mathbb{P}_{\mathrm{del}}) \leq \sqrt{\tfrac{1}{2}\mathrm{KL}(\mathbb{P}_{\mathrm{del}}\|\pi^\star)}. \tag{40}$$

Therefore

$$d_{\mathrm{TV}}(\pi^\star, \mathbb{P}_{\mathrm{del}}) \leq \sqrt{\frac{1}{2}\left(\mathrm{KL}(\mathbb{P}_{\mathrm{del}}\|p_0) + \frac{\mathbb{E}_{p_0}[s] - \mathbb{E}_{\mathbb{P}_{\mathrm{del}}}[s]}{\tau} + \frac{1}{8\tau^2}\right)}. \tag{41}$$

**Relating separability to the bias.** By Definition 2 (Trace separability), $\mathbb{E}_{\mathbb{P}_{\mathrm{del}}}[s] \geq \mathbb{E}_{\mathbb{P}_{\mathrm{ret}}}[s] + \Delta$. If $p_0$ is a public prior independent of the unlearning operation, we can treat $\mathbb{E}_{p_0}[s]$ as a constant that does not increase with the separability margin; hence the difference $\mathbb{E}_{p_0}[s] - \mathbb{E}_{\mathbb{P}_{\mathrm{del}}}[s]$ decreases as $\Delta$ grows, which tightens equation 41. Consequently, the total variation bias can be controlled by $(\tau, \Delta)$ and $\mathrm{KL}(\mathbb{P}_{\mathrm{del}}\|p_0)$.

**Step 4: From TV to the chosen metric $d(\cdot, \cdot)$.** On a bounded domain $\mathcal{X}$ with diameter $D$,

$$W_1(\pi^\star, \mathbb{P}_{\mathrm{del}}) \leq D \cdot d_{\mathrm{TV}}(\pi^\star, \mathbb{P}_{\mathrm{del}}). \tag{42}$$

For MMD with kernel $k$ bounded by $K$,

$$\begin{aligned}
\mathrm{MMD}(\pi^\star, \mathbb{P}_{\mathrm{del}}) &= \sup_{\|f\|_{\mathcal{H}}\leq 1} \left| \mathbb{E}_{\pi^\star}[f] - \mathbb{E}_{\mathbb{P}_{\mathrm{del}}}[f] \right| \\
&\leq \sup_{\|f\|_{\mathcal{H}}\leq 1} \|f\|_\infty \cdot 2\,d_{\mathrm{TV}}(\pi^\star, \mathbb{P}_{\mathrm{del}}) \\
&\leq 2\sqrt{K}\,d_{\mathrm{TV}}(\pi^\star, \mathbb{P}_{\mathrm{del}}).
\end{aligned} \tag{43}$$

using $\|f\|_\infty \leq \sqrt{K}\|f\|_{\mathcal{H}}$. More generally, for any IPM induced by a function class $\mathcal{F}$ with $\|f\|_\infty \leq B$,

$$d(\pi^\star, \mathbb{P}_{\mathrm{del}}) \leq 2B\,d_{\mathrm{TV}}(\pi^\star, \mathbb{P}_{\mathrm{del}}). \tag{44}$$

Combine with equation 41 to obtain an explicit bias bound:

$$d(\pi^\star, \mathbb{P}_{\text{del}}) \leq C_{\text{met}} \cdot \sqrt{\tfrac{1}{2}\left(\text{KL}(\mathbb{P}_{\text{del}}\|p_0) + \tfrac{\mathbb{E}_{p_0}[s] - \mathbb{E}_{\mathbb{P}_{\text{del}}}[s]}{\tau} + \tfrac{1}{8\tau^2}\right)}, \tag{45}$$

$$\text{where } C_{\text{met}} = \begin{cases} D, & d = W_1, \\ 2\sqrt{K}, & d = \text{MMD}, \\ 2B, & \text{general } d. \end{cases}$$

Define $C_1(\tau, \Delta)$ as the right-hand side; it decreases as $\tau \downarrow 0$ and as the separability margin $\Delta$ increases (since $\mathbb{E}_{\mathbb{P}_{\text{del}}}[s]$ increases with $\Delta$).

**Step 5: Put together**. With probability at least $1 - \delta$, we have

$$d(\widetilde{P}_k, \pi^\star) \leq \epsilon(k, \delta), \tag{46}$$

where $\epsilon(k, \delta)$ is the sampling error of Step 2 (e.g., $2\sqrt{K \log(2/\delta)/k}$ for MMD, or $C_d(\log(2/\delta)/k)^{1/d}$ for $W_1$). Hence, by equation 31 and the stability of $\mathcal{R}$,

$$d(\widehat{P}_k, \mathbb{P}_{\text{del}}) \leq C_L \epsilon(k, \delta) + C_1(\tau, \Delta), \tag{47}$$

which proves the theorem. □

## D EVALUATION

### D.1 DETAILED EXPERIMENTAL SETUP

**Unlearning method**. Following existing work (Hu et al., 2024), to obtain the paired models $(f^+, f^-)$, we adopt two categories of unlearning procedures. *Exact unlearning* is implemented by removing the forgotten samples and fine-tuning the model on the remaining data for the same number of epochs as the original training. *Approximate unlearning* is implemented using the single gradient unlearning method (Thudi et al., 2022). In both cases, unlearning is performed in a *class-wise* manner, which reflects realistic scenarios where requests often target all samples belonging to a specific semantic category. We set class 0 as the default target for unlearning.

**Datasets**. We evaluate on CIFAR-100 with 50k training images and 10k test images of size $32 \times 32$ RGB. This dataset contains 100 object categories with relatively low resolution and high intra-class variability, making it a standard benchmark for image classification. Food-101 includes 75k training images and 25k test images of size $224 \times 224$ RGB. It covers 101 food categories collected from real-world scenarios, characterized by large intra-class diversity, occlusion, and noisy labels. PathMNIST, a subset of the MedMNIST collection, consists of 89,996 training images, 7,180 validation images, and 7,180 test images of size $28 \times 28$ grayscale. It provides 9 classes derived from colorectal cancer histology slides, capturing diverse tissue types such as adipose, lymphocytes, smooth muscle, and adenocarcinoma epithelium.

**Models** ($f^+, f^-$). We adopt ResNet-18 on CIFAR-100, Food-101, and PathMNIST datasets for image classification tasks. We use cross-entropy loss with the SGD optimizer and a cosine learning rate schedule, and maintain a checkpoint pair $(f^+, f^-)$ for each dataset.

**Generator (RL Policy $\pi_\phi$)**. We use a DCGAN-style generator $G$ that maps a latent vector $z \in \mathbb{R}^d$ to $32 \times 32$ images with $\tanh$ outputs in $[-1, 1]$, and an RL policy $\pi_\phi$ that produces $z$ via a diagonal Gaussian whose mean and log-std are predicted by a two-layer MLP. At each step, the policy samples $z = \mu_\phi(\epsilon) + \exp(\log \sigma_\phi(\epsilon)) \odot \epsilon$ with $\epsilon \sim \mathcal{N}(0, I)$, generates $x_{\text{fake}} = G(z)$, and receives a reward built from unlearning traces between $f^+$ and $f^-$. The policy is optimized with PPO (clipped objective with entropy bonus), while $G$ can be jointly fine-tuned by a surrogate loss that maximizes the trace signal and includes teacher-guided KL distillation from $f^+$ and $f^-$, and total-variation regularization. In practice, we initialize $G$ with different pretrained GANs depending on the dataset: an ImageNet-pretrained GAN for CIFAR-100 and Food-101, and a MedMNIST-pretrained GAN for PathMNIST.

**RETRACE settings**. We set $\alpha = 0.4$, $\beta = 0.2$, and $\gamma = 0.4$ as the relative weights for trace components. For reconstruction, we select samples using top-$k$ with $k = 64$.

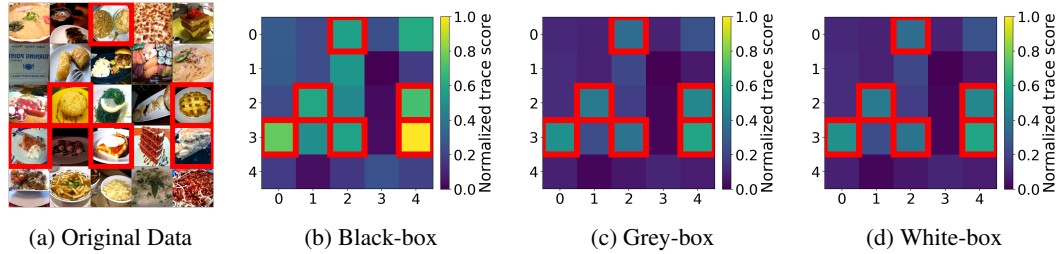

Figure 7: Traces of unlearned data under different model access levels on Food-101 in the approximate unlearning scenario. The images in red boxes represent the unlearned data "*Apple pie*".

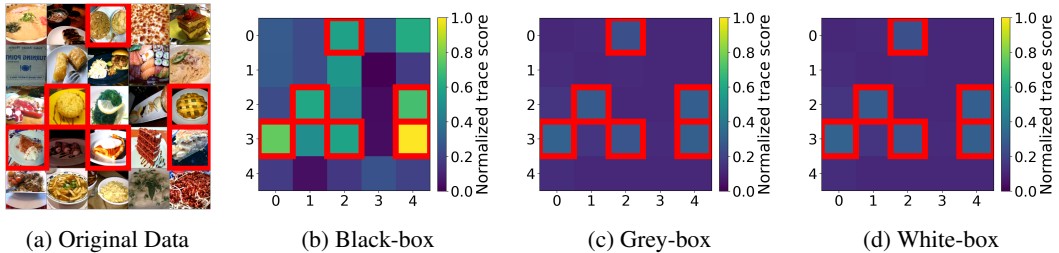

Figure 8: The normed version of Figure 7.

**Evaluation metrics**. For instance-level evaluation, we use *Mean Squared Error (MSE)* to assess the *pixel-level* difference between two images; *cosine similarity (CS)* to evaluate the *feature-level* difference; *success rate (SR)* to evaluate the percentage of generated images being unlearned class. For distribution-level evaluation, we use *Fréchet Inception Distance (FID)* to evaluate the visual and *statistical similarity* between reconstructed and original distributions, and *Kullback–Leibler (KL) divergence* to assess the alignment of their *probability distributions*.

**Baseline methods**. We compare RETRACE with two state-of-the-art reconstruction attack methods, which are listed as follows.

- **Unlearning Inversion Attack (UIA)** (Hu et al., 2024), which is conducted to recover unlearned data in white-box access.
- **HRec** (Bertran et al., 2024), which achieves nearly-perfect attack on linear regression and can be generalized to other model architectures.

## D.2 TRACE VISUALIZATION

For both Food-101 and PathMNIST, we present the $f^+ - f^-$ trace heatmaps and the corresponding trace distributions (Figures 7, 9, 10, 12).

## D.3 EFFECTIVENESS OF RETRACE

**Instance level**. Table 1 presents the instance-level reconstruction results of RETRACE across three datasets and three model-access levels. For the criteria of MSE and CS, we compute all pairwise distances within the unlearned class in the original dataset and take the average, which reflects the natural intra-class variability (values are presented in tablenote).

Under the *exact unlearning* setting, across three datasets, the reconstructed samples achieve MSE values that are only slightly higher than the intra-class criteria (e.g., CIFAR-100: 0.20 vs. 0.16; Food-101: 0.23 vs. 0.13), showing that the pixel-level differences between reconstructions and unlearned images remain small. The SR values are consistently around 50–60%, indicating that our method can successfully generate a large proportion of samples that are classified as belonging to the unlearned class. Meanwhile, the CS values (e.g., CIFAR-100: 0.47 vs. 0.57) demonstrate that the reconstructed

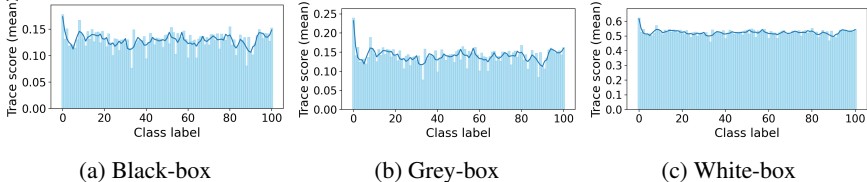

(a) Black-box        (b) Grey-box        (c) White-box

Figure 9: Trace distribution under different model access levels on Food-101 in the exact unlearning scenario. Class 0 represents "*Apple pie*".

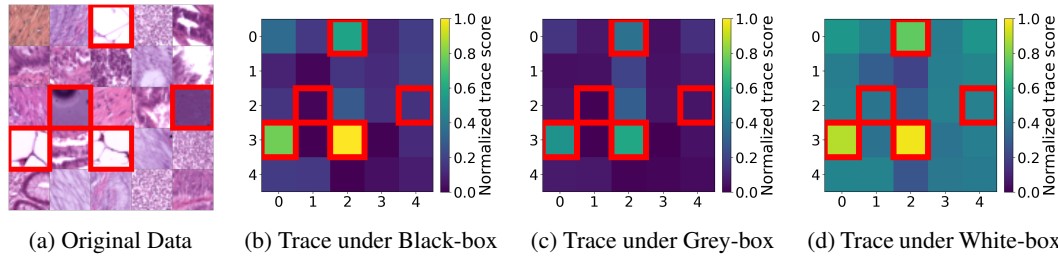

(a) Original Data    (b) Trace under Black-box    (c) Trace under Grey-box    (d) Trace under White-box

Figure 10: Traces of unlearned data under different model access levels on PathMNIST in the approx- imate unlearning scenario. The images in red boxes represent the unlearned data "*Adipose*".

samples lie close to the original unlearned data in feature space, confirming that the recovered images retain strong semantic similarity.

In the *approximate unlearning* scenario, the performance further improves across datasets and access levels. MSE decreases noticeably compared to exact unlearning (e.g., CIFAR-100: 0.17 vs. 0.20), suggesting that approximate unlearning leaves stronger residual signals for pixel-level recovery. SR also increases significantly, with white-box access achieving the highest rates (e.g., CIFAR-100: 73.1%; PathMNIST: 59.7%), demonstrating that our method can generate an even larger number of valid samples for the forgotten class. Finally, CS values are consistently close to the criteria (e.g., Food-101: 0.49 vs. 0.54), indicating that reconstructed images not only recover visual details but also align well with the semantic representations of unlearned data.

**Distribution level**. Figure 6 illustrates the distribution-level reconstruction results of RETRACE in terms of FID and KL divergence.

Under exact unlearning, across all three datasets, RETRACE achieves meaningful reconstruction performance under all access levels. In the black-box setting, both FID and KL remain at reasonably low values (e.g., CIFAR-100: FID = 142.1, KL = 3.62; PathMNIST: FID = 125.4, KL = 3.33), showing that the method can approximate the deleted distribution even with limited information. Moving to the grey-box setting, the incorporation of loss-level traces consistently reduces both FID and KL (e.g., CIFAR-100: FID drops from 142.1 to 135.0, KL from 3.62 to 2.55), demonstrating improved alignment with the original distribution. The best results are obtained in the white-box setting, where gradient-level traces further enhance reconstruction quality, yielding the lowest FID and KL across datasets (e.g., CIFAR-100: FID = 108.9, KL = 2.49; PathMNIST: FID = 96.5, KL = 2.22). These results highlight that while RETRACE is effective even under restrictive black-box conditions, more informative access significantly boosts distribution-level recovery.

In the approximate unlearning setting, the results further improved. Compared to exact unlearning, FID values are consistently lower (e.g., CIFAR-100: 99.1 vs. 108.9; PathMNIST: 93.4 vs. 96.5), reflecting that approximate unlearning leaves stronger distributional traces that RETRACE can exploit. KL divergence also decreases slightly in most cases (e.g., PathMNIST: 2.00 vs. 2.22), confirming that the reconstructed samples align closely with the underlying class distribution.

**Cases**. We present reconstructed samples at the *instance level* for CIFAR-100, Food-101, and PathMNIST under the white-box setting.

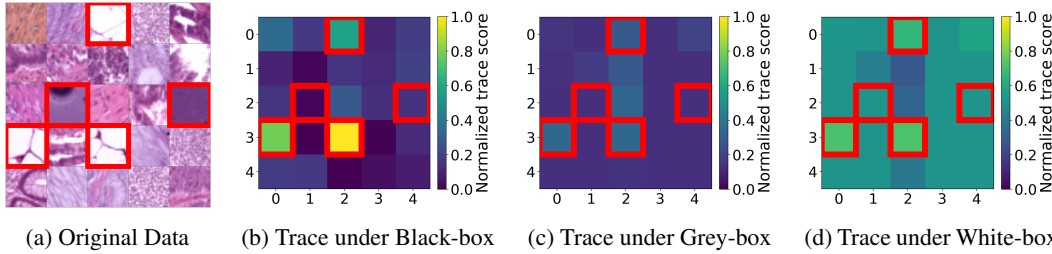

| (a) Original Data | (b) Trace under Black-box | (c) Trace under Grey-box | (d) Trace under White-box |

Figure 11: The normed version of Figure 10.

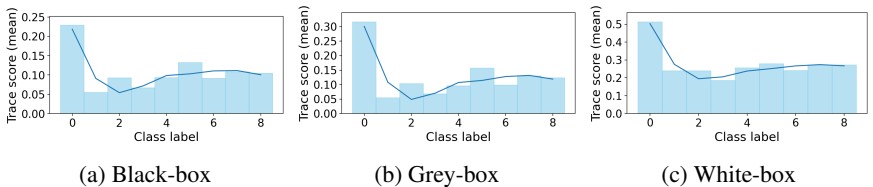

| (a) Black-box | (b) Grey-box | (c) White-box |

Figure 12: Trace distribution under different model access levels on PathMNIST in the exact unlearning scenario. Class 0 represents "*Adipose*".

For CIFAR-100, we present reconstructed samples under *Exact Unlearning* (Figure 13).

For Food-101, we present reconstructed samples under *Exact Unlearning* (Figure 15) and under *Approximate Unlearning* (Figure 14).

For PathMNIST, we present reconstructed samples under *Exact Unlearning* (Figure 17) and under *Approximate Unlearning* (Figure 16).

### D.4 DISTRIBUTION-LEVEL RECONSTRUCTION RESULTS OF BASELINES

To make the results more convincing on evaluating RETRACE's effectiveness, we further plot the distribution-level reconstruction ability of two baseline methods. Table 6 provides the results regarding FID and KL values of them. Compared with their performance, RETRACE performs (Figure 6) much better than both UIA and HRec. Across all three datasets, and under both exact and approximate unlearning, RETRACE consistently achieves the lowest FID and KL divergence, outperforming UIA and HRec. While the baselines yield relatively large FID scores (typically above 200 on CIFAR-100 and Food-101, and above 140 on PathMNIST) and KL values around 4–6, RETRACE substantially reduces both metrics across all threat models. As shown in Figure 6, our method achieves markedly lower FID (e.g., 99–142 on CIFAR-100 and 93–125 on PathMNIST) and KL (approximately 2–4) in Black-box, Grey-box, and White-box settings. This consistent improvement demonstrates that RETRACE reconstructs samples on distribution-level that align much more closely with the deleted data distribution than existing baselines.

### D.5 RETRACE MULTI-RUN RESULTS WITH STANDARD DEVIATION

We present the RETRACE of multi-run results with standard deviation in Table 7. Our main evaluation spans three datasets of different scales and domains, and across all of them, the white-box setting consistently achieves reliably stronger reconstruction performance. This consistency across heterogeneous tasks indicates that the observed improvements are not accidental fluctuations, but reflect the true behavior of the method under increased model access level. In our evaluation, we ran 5 independent experiments, so the results presented in Table 1 reflect the mean of the 5 runs. Thus, we further provide the standard deviation across these runs to confirm that the observed gains are not due to randomness but remain statistically robust. The results confirm that the improvements in the white-box setting persist under multi-run evaluation. Therefore, the white-box advantage is statistically stable rather than due to fluctuations.

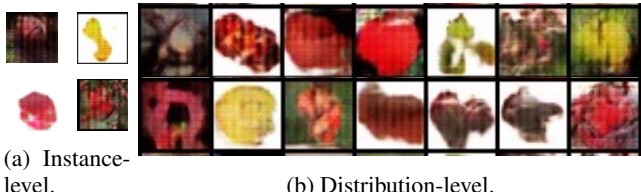

(a) Instance-level.

(b) Distribution-level.

Figure 13: Generated unlearned samples ("Apple class") in CIFAR-100 under exact unlearning.

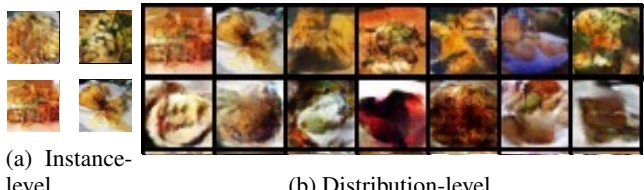

(a) Instance-level.

(b) Distribution-level.

Figure 14: Generated unlearned samples ("Apple pie class") in Food-101 under approximate unlearning.

### D.6   RETRACE UNDER ADDITIONAL UNLEARNING METHODS

We provide the RETRACE result under additional three unlearning methods in Table 8.

### D.7   ABLATION ON UNLEARNED CLASSES

We present a group of examples on *reconstructed samples* for multiple CIFAR-100 classes under *Exact* and *Approximate* unlearning, as shown in Figure 18.

### D.8   SCALABILITY ON RESNET-34

We evaluate the scalability of RETRACE by scaling the neural network up to Resnet-34 on CIFAR100 dataset. The results are presented in Figure 19.

### D.9   RETRACE'S PERFORMANCE ON UNSEEN CLASSES

Table 9 lists the performance of RETRACE on reconstructing unseen classes.

### D.10   GENERALIZATION ON TEXT DATASET

**Setup**. We employ a pretrained GPT-2 (small) decoder as the text generator and fine-tune it with a lightweight PPO/REINFORCE objective, while the DistilBERT classifiers (Sanh et al., 2019) serve as unlearned model: $f^+$ trained on the full AG News corpus and $f^-$ retrained after *Exact Unlearning* of the unlearned class. At reconstruction time, we query $f^+$ and $f^-$ on sampled texts to construct a *trace* reward whose components depend on the access setting: Black-box uses the prediction divergence $\delta_{\text{pred}}(x) = \|\text{softmax}(f^+(x)) - \text{softmax}(f^-(x))\|_2$; Gray-box augments this with a loss gap $\delta_{\text{loss}}(x) = \left|\ell(f^+(x), \hat{y}) - \ell(f^-(x), \hat{y})\right|$ where $\hat{y} = \arg\max f^+(x)$; White-box additionally incorporates a representation discrepancy $\delta_{\text{feat}}(x) = 1 - \cos\left(h^+(x), h^-(x)\right)$ from hidden states. The total reward is the weighted sum of the trace term, a class-prior from $f^+$ (target-class probability), a discriminative term that increases the target-class logit, a small fluency bonus from the policy log-likelihood, and a length penalty. We decode with top-$k$ sampling, a repetition penalty, and a minimum new-token budget to mitigate collapse. We evaluate reconstruction effectiveness with BLEU, computed as corpus-level $n$-gram overlap against held-out texts from the forgotten class, and MMD, computed between generated and forgotten-class distributions in the DistilBERT [CLS] embedding space using a multi-bandwidth RBF kernel.

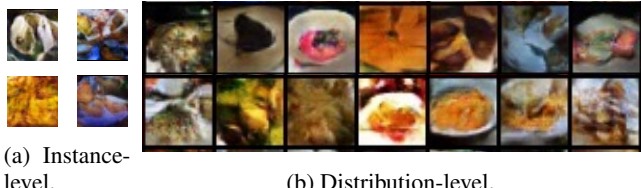

(a) Instance-
level.

(b) Distribution-level.

Figure 15: Generated unlearned samples ("Apple pie class") in Food-101 under exact unlearning.

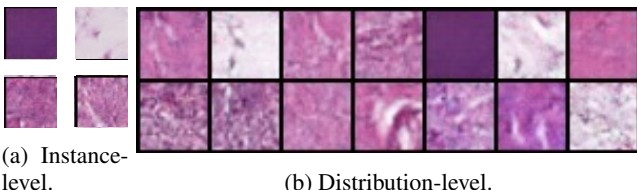

(a) Instance-
level.

(b) Distribution-level.

Figure 16: Generated unlearned samples ("Adipose class") in PathMNIST under approximate un-
learning.

**Reconstruction examples on the text dataset**. The representative reconstructions on AG News
(forgotten class: *Sports*) are shown below. These examples illustrate the recovered sports style.

**Sample 0:** Inter lift Coppa Italia after 1–0 final; Martínez scores decisive header.
**Sample 1:** Celtics survive Heat 104–99; Tatum posts 34–9–7 in Game 5.
**Sample 2:** France edge Spain 1–0 to lift UEFA Nations League crown in Milan.

### D.11 DISCUSSION OF POTENTIAL LIMITATIONS

Our method builds upon pretrained generative models, which serve as the base model for recon-
structing the unlearned content. This design alleviates the challenges of training from scratch, such
as unstable optimization and mode collapse, and enables more efficient adaptation with RL signals.
Nevertheless, the characteristics of the pretrained model itself naturally influence the reconstruc-
tion quality. In particular, the alignment between the pretraining corpus and the target task domain
plays a critical role: if the pretraining data diverges significantly from the forgotten distribution, the
generated samples may deviate from the intended semantics or result in unsatisfactory outputs.

However, in modern machine learning practice, the use of pretrained models has become standard.
Large-scale pretraining not only reduces computational overhead but also greatly improves practi-
cality compared to training from scratch. Therefore, this potential limitation, which could otherwise
affect RETRACE's performance, is unlikely to pose a significant concern.

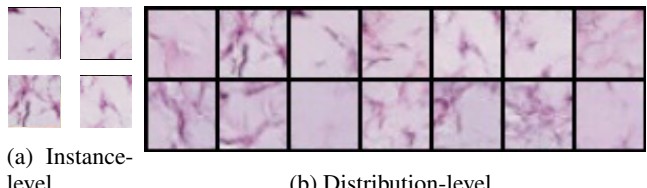

(a) Instance-level.

(b) Distribution-level.

Figure 17: Generated unlearned samples ("Adipose class") in PathMNIST under exact unlearning.

Table 6: FID and KL divergence comparison under baselines.

| Dataset | Method | Exact Unlearning | | Approximate Unlearning | |
|---|---|---|---|---|---|
| | | FID | KL | FID | KL |
| CIFAR-100 | UIA | 203.6 | 5.2 | 189.7 | 4.8 |
| | HRec | 230.4 | 5.3 | 201.4 | 4.9 |
| Food-101 | UIA | 224.8 | 5.6 | 212.3 | 5.1 |
| | HRec | 241.6 | 5.3 | 226.6 | 5.1 |
| PathMNIST | UIA | 156.4 | 4.1 | 143.4 | 3.8 |
| | HRec | 148.2 | 4.3 | 139.4 | 4.0 |

Table 7: Mean and standard deviation of 5 independent runs.

| | **Exact Unlearning** | | | | | | | | |
|---|---|---|---|---|---|---|---|---|---|
| Dataset | Black-box | | | Grey-box | | | White-box | | |
| | MSE | SR | CS | MSE | SR | CS | MSE | SR | CS |
| CIFAR-100 | 0.23±0.05 | 52.3%±0.4% | 0.43±0.04 | 0.22±0.03 | 58.7%±0.5% | 0.46±0.03 | 0.20±0.02 | 62.4%±0.6% | 0.47±0.02 |
| Food-101 | 0.25±0.04 | 48.9%±0.5% | 0.38±0.04 | 0.23±0.03 | 54.7%±0.4% | 0.42±0.02 | 0.23±0.02 | 50.4%±0.6% | 0.44±0.02 |
| PathMNIST | 0.26±0.04 | 49.8%±0.5% | 0.26±0.03 | 0.23±0.02 | 50.6%±0.4% | 0.27±0.02 | 0.22±0.01 | 52.8%±0.5% | 0.31±0.02 |

| | **Approximate Unlearning** | | | | | | | | |
|---|---|---|---|---|---|---|---|---|---|
| Dataset | Black-box | | | Grey-box | | | White-box | | |
| | MSE | SR | CS | MSE | SR | CS | MSE | SR | CS |
| CIFAR-100 | 0.24±0.04 | 55.9%±0.6% | 0.46±0.03 | 0.21±0.03 | 68.7%±0.4% | 0.49±0.03 | 0.17±0.02 | 73.1%±0.5% | 0.50±0.02 |
| Food-101 | 0.24±0.03 | 56.6%±0.5% | 0.41±0.03 | 0.18±0.02 | 62.5%±0.5% | 0.49±0.02 | 0.16±0.01 | 65.3%±0.4% | 0.49±0.01 |
| PathMNIST | 0.24±0.03 | 51.9%±0.5% | 0.28±0.02 | 0.20±0.02 | 54.6%±0.5% | 0.28±0.02 | 0.19±0.02 | 59.7%±0.5% | 0.33±0.01 |

Table 8: RETRACE under additional unlearning methods.

| Method | Black-box | | | Grey-box | | | White-box | | |
|---|---|---|---|---|---|---|---|---|---|
| | MSE | SR | CS | MSE | SR | CS | MSE | SR | CS |
| Sparsity-based Unlearning | 0.21 | 53.1% | 0.46 | 0.20 | 63.8% | 0.47 | 0.18 | 70.9% | 0.49 |
| Boundary Unlearning | 0.20 | 53.5% | 0.47 | 0.19 | 65.2% | 0.49 | 0.17 | 71.3% | 0.51 |
| Convolution-Transpose Unlearning | 0.22 | 53.4% | 0.46 | 0.21 | 63.2% | 0.49 | 0.18 | 73.4% | 0.50 |

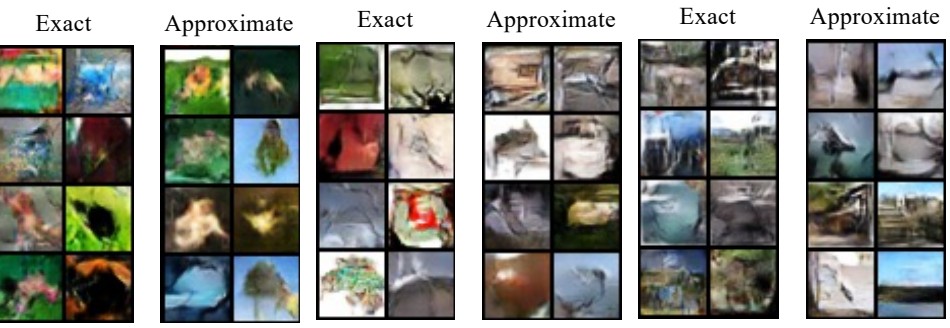

Exact  Approximate  Exact  Approximate  Exact  Approximate

(a) Class 1: Aquarium fish.    (b) Class 5: Bed.    (c) Class 12: Bridge.

Figure 18: Reconstructed unlearned samples on class 1, 5 and 12 on CIFAR-100.

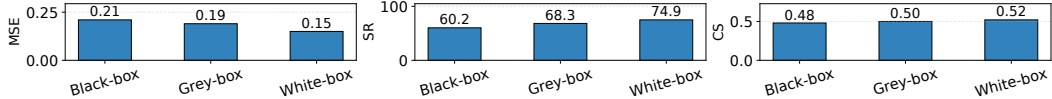

Figure 19: The performance of RETRACE on CIFAR100 when the target model is scaled up to a ResNet-34

Table 9: RETRACE's performance on unseen class.

| Method | Access | MSE | SR | CS | FID | KL |
|--------|--------|------|-------|------|-------|-----|
| Exact | BB | 0.33 | 42.1% | 0.32 | 203.4 | 5.3 |
| | GB | 0.30 | 43.7% | 0.36 | 199.2 | 4.9 |
| | WB | 0.31 | 45.4% | 0.38 | 183.8 | 4.7 |
| Approx. | BB | 0.27 | 51.3% | 0.42 | 182.1 | 4.2 |
| | GB | 0.27 | 58.5% | 0.43 | 169.7 | 4.0 |
| | WB | 0.24 | 59.6% | 0.45 | 158.9 | 3.7 |

