# OpenReview forum: "ReTrace: Reinforcement Learning-Guided Reconstruction Attacks on Machine Unlearning"
_ICLR.cc/2026/Conference — ICLR 2026 Poster_

### Official Review · Reviewer_tMmg · 2025-10-30

**Soundness:** 3
**Presentation:** 3
**Contribution:** 2
**Rating:** 4
**Confidence:** 3

**Summary:**

This paper proposes ReTrace, which seeks to reconstruct unlcearned training samples assuming access to the model before and after unlearning. ReTrace uses a reinforcement learning to learn a sample that is likely to minimize the loss functions of the two models before and after unlearning.

**Strengths:**

Understanding and objectively measuring the effectiveness of unlearning is an important problem. The paper presents a novel and interesting use of RL in this space. Overall, I think as a measurement method, there are some benefits to ReTrace as the results seem to suggest that exact unlearning is more effective than approximate unlearning, which is a nice, though somewhat expected result.

**Weaknesses:**

The paper claims to evaluate both ResNet and DistilBERT, but the majorit of results are in the image domain. Average attack success rates is only around 50% for exact unlearning and 60% for approximate unlearning, and worse on the text tasks, so the attack is not that  effective. As an actual attack, ReTrace doesn't seem that realistic: as it assumes that an adversary can access both pre- and post-unlearning models while running a costly RL process. The paper would have been stronger if the authors could have further compared the relative strenghts of different unlearning proposals using ReTrace and used their method to explain some of the differences.

**Questions:**

I found some results confusing: some trace score patterns in Figure 2 and Figure 6 are confusing as the trace score doesn't necessarily correlate with being unldearned or not —for example, (0,0) has a very low trace score despite being unlearned, while other points like (1,0) and (1,1) show high scores even though they are not unlearned. The bottom-right case (4,4) is inconsistent across black-, gray-, and white-box settings -- an explanation would be nice.

---

> ### Author Response · Authors · 2025-11-23
> **Response to Reviewer tMmg (1/2)**
>
> We thank the reviewer for the insightful and constructive comments. We are confident to address all the raised weaknesses and questions.
>
> **W1. Evaluation results in text domain**.
> Our main experiments (Sections 5.1 and 5.3) focus on vision datasets because image classification remains the dominant benchmark in machine unlearning and enables direct comparison with prior reconstruction attacks. However, ReTrace is not vision-specific, in Section 5.4, we explicitly demonstrate its applicability to NLP by evaluating on DistilBERT, which to the best of our knowledge is the largest architecture attacked for unlearning reconstruction to date. The results confirm that ReTrace generalizes beyond ResNet and CNNs, successfully extracting forgotten textual content.
>
> **W2. The effectiveness of ReTrace regarding the attack success rate**.
> We thank the reviewer for pointing out the success rate of ReTrace. We evaluate the effectiveness of ReTrace under three model access levels (black-, gray-, and white-box settings), reflecting the real-world attack scenarios. Our obtained success rate in Table 1 and 2 (Section 5.3) indeed shows the effectiveness of ReTrace. Instance-level reconstruction in machine unlearning is an inherently challenging, high-dimensional generation task, and success rates in the around 50%–70% range, achieved across three datasets and model access settings, represent strong attack performance. In the black-box setting, existing update-based unlearning attacks in prior work typically struggle or fail due to they need full model structure and parameters, but ReTrace can still achieve around 50% success rates, substantially outperforming existing baselines. In the white-box setting, baseline methods can execute the attack based on their approach (this is also the reason that we only report the comparison of ReTrace and baselines under white box setting), but ReTrace significantly performs better according to Table 2, increasing the success rate by around 20%. These results show the attack efficacy rather than marginal improvement of ReTrace.
>
> **W3. Assumption of accessing to $f^+$ and $f^-$**.
> We thank the reviewer for pointing out this. In fact, our attack is realistic, as the assumption of accessing both original model and unlearned model is a commonly occurring situation in practice. We have already discussed the scenarios in Section 2.2 and Appendix A. The details are summarized as follows.
> - (API versioning). OpenAI, Google Cloud Vision, Azure Cognitive Services simultaneously expose multiple model versions (e.g., preview vs stable), allowing queries to both old and updated models.
> - (Canary rollouts). Meta, Google, and other large-scale platforms run or release old and new model versions concurrently for months during deployment.
> - (Rollback requirements in regulated industries). Finance/healthcare regulations require that previous checkpoints be preserved and auditable, making f^+ accessible through insider risk.
> Thus our assumption in the threat model is reasonable, and it reflects how real systems operate.
>
> We also encourage the reviewer to see our response to the weakness 4 of Reviewer FHN1.
>
> **W4. Computational overhead**.
> The computational cost of ReTrace mainly consists of three aspects: (1) Trace extraction; (2) Policy updates via PPO; (3) Reconstruction step, which is elaborated in Section 3.4 and Appendix B. The complexity simplifies to $O(N+NlogN)$, which is comparable to or lower than existing reconstruction attacks such as gradient ascent, latent inversion, or diffusion inversion.
>
> We refer the reviewer to our response to reviewer gE9i – W3.

---

> ### Author Response · Authors · 2025-11-23
> **Response to Reviewer tMmg (2/2)**
>
> **W5. Test on more unlearning methods**.
> We thank the reviewer for this constructive suggestion. Following the comment, we additionally applied ReTrace to three representative unlearning baselines, Influence Unlearning (sparsity-based) [1], Boundary Unlearning [3], and Convolution-Transpose Unlearning [2]. The results are listed as follows. We encourage the reviewer to see our response to Q7 of Reviewer FHN1 and W1 of Reviewer gE9i as well for this point.
>
> | Method                          | Black-box MSE  | Black-box SR  | Black-box CS  | Grey-box MSE  | Grey-box SR  | Grey-box CS  | White-box MSE  | White-box SR  | White-box CS  |
> |---------------------------------|-----------------|----------------|----------------|----------------|----------------|----------------|------------------|-----------------|----------------|
> | Sparsity-based Unlearning       | 0.21          | 53.1%          | 0.46           | 0.20         | 63.8%          | 0.47           | 0.18           | 70.9%           | 0.49           |
> | Boundary Unlearning             | 0.20          | 53.5%          | 0.47          | 0.19         | 65.2%          | 0.49          | 0.17           | 71.3%           | 0.51           |
> | Convolution-Transpose Unlearning| 0.22          | 53.4%          | 0.46          | 0.21         | 63.2%          | 0.49          | 0.18           | 73.4%           | 0.50          |
>
>
> **Q1. Trace visualization in Figures 2 and 6**.
> We thank the reviewer for this careful observation. We clarify that the visualizations in Figures 2 and 6 are not intended to show that every unlearned sample always has the highest absolute score, but to illustrate how the separability between unlearned vs retained data improves as the model access level increases (black $\rightarrow$ grey $\rightarrow$ white).
> - Figures 2 and 6 compare trace patterns of unlearned vs retained samples under three access levels. Across both figures, a consistent phenomenon emerges. Overall, in both the grey-box and white-box settings, retained (non-unlearned) samples exhibit more concentrated trace distributions with smaller variance compared to the black-box setting. Unlearned samples similarly show tighter concentration, lower variance, and greater stability. This trend aligns with our experimental findings: richer adversarial information in grey-box and white-box settings leads to more stable trace signatures and higher reconstruction success rates.
> - Some retained samples show unusually high trace scores (e.g., (1,0), (1,1)) and some unlearned samples exhibit lower scores (e.g., (0,0)) in Figure 6. This behavior is expected and arises naturally from dataset-inherent noise; class overlapping; samples near decision boundaries; or natural outliers. These anomalous points are much more visible in the black-box setting, where the score signal is weak and unstable.
> As the reviewer also noticed, these inconsistencies diminish in the grey- and white-box settings, because richer access to model internals stabilizes the trace signals. This is also the phenomenon our visualizations aim to reveal.
> - To improve clarity, we will add normalized visualizations. We have generated a normalized version of Figure 2, where for each 5×5 patch we compute the patch-average trace value and the ratio between each highlighted red-square value and the patch average. In this normalized visualization, white-box shows the highest ratio gap between unlearned and retained data; grey-box shows a moderate ratio, and black-box shows weak but acceptable ratios, fully matching our quantitative evaluation. We will include this normalized version in the revision.
> We also refer the reviewer to our response to weakness 7 (W7) of Reviewer FHN1, where we further elaborate the trace distribution behavior.
>
> **Reference**
>
> [1] Thudi, A., Deza, G., Chandrasekaran, V., & Papernot, N. Unrolling sgd: Understanding factors influencing machine unlearning. In 2022 IEEE 7th European Symposium on Security and Privacy (EuroS&P) (pp. 303-319). IEEE.]
>
> [2] Cadet, X. F., Borovykh, A., Malekzadeh, M., Ahmadi-Abhari, S., & Haddadi, H. (2025, June). Deep Unlearn: Benchmarking Machine Unlearning for Image Classification. In 2025 IEEE 10th European Symposium on Security and Privacy (EuroS&P) (pp. 939-962). IEEE.
>
> [3] Chen, M., Gao, W., Liu, G., Peng, K., & Wang, C. (2023). Boundary unlearning: Rapid forgetting of deep networks via shifting the decision boundary. In Proceedings of the IEEE/CVF Conference on Computer Vision and Pattern Recognition (pp. 7766-7775).

---

### Official Review · Reviewer_MuvD · 2025-11-01

**Soundness:** 3
**Presentation:** 2
**Contribution:** 4
**Rating:** 6
**Confidence:** 3

**Summary:**

This paper presents ReTrace, a new reconstruction attack that reinterprets the problem of recovering unlearned data through the lens of reinforcement learning (RL). Instead of relying on static optimization or inversion techniques, ReTrace uses discrepancies between the original and unlearned models as reward signals to guide a pretrained generator toward regions of the input space that likely correspond to forgotten data. The approach integrates multiple trace signals—changes in predictions, losses, and gradients—across different access levels (black-, grey-, and white-box) and performs reconstruction through RL-guided latent exploration followed by a candidate refinement step. Through both theoretical analysis and experiments on multiple datasets, the paper shows that ReTrace can recover semantically meaningful data at both the instance and distribution levels, highlighting residual vulnerabilities in existing unlearning methods.

**Strengths:**

- **Novel and creative formulation**. This paper presents a genuinely original framing of reconstruction attacks. It adapts the RL-GAN-Net idea from Sarmad et al. (2019)—originally proposed for conditional image generation—and re-purposes it for unlearning data recovery. Using reinforcement learning to guide a generator’s latent exploration based on unlearning traces is both conceptually interesting and technically innovative.

- **General and modular framework**. The approach unifies different attack settings (black-, grey-, and white-box) under a single reward formulation, where prediction, loss, and gradient discrepancies can be combined or omitted depending on model access. This modularity makes the framework broadly applicable and easy to adapt to new unlearning scenarios.

- **Empirical evidence of residual traces after unlearning**. The experiments clearly demonstrate that models subjected to unlearning still leak identifiable information, confirming the practical relevance of the attack.

- **Largest-scale model targeted to date**. The paper extends reconstruction attacks beyond CNNs to transformer architectures, reporting results on DistilBERT, which the authors state is the largest model attacked to date in the unlearning literature. This demonstrates the framework’s scalability and suggests that the proposed RL-based formulation might have the potential to generalize to high-dimensional transformer settings.

- **Timely and relevant contribution**. The work addresses the emerging area of machine-unlearning security—a topic of growing importance for model safety, privacy, and regulatory compliance—and provides a concrete framework for analyzing these vulnerabilities.

- **Theoretical grounding supporting intuition**. The paper includes a concise theoretical analysis that explains why the proposed RL formulation works, strengthening the intuition behind the method.

**Weaknesses:**

I like this paper — it’s a creative and well-motivated idea. That said, there are a few points I would like to raise and hopefully discuss with the authors.

- **On the ambiguity in the RL formulation**. Section 3.2 introduces the RL framing with definitions of state, action, and policy, but these descriptions are somewhat abstract and internally inconsistent when mapped to the actual image-generation setup. The text defines the action as “sampling or refining a candidate x from the generator,” implying that the policy acts in the data space, while simultaneously describing the policy πϕ as “outputting candidates from the latent space,” implying it acts over z. This leaves it unclear whether the RL agent’s action space is x or z. Appendix D.1 later reveals that πϕ is in fact a small two-layer MLP producing latent vectors z that are then passed through a pretrained DCGAN G to obtain x = G(z). This architectural detail is crucial for understanding the proposed RL loop but is only specified in the appendix under the Experimental Setup Section. I would strongly recommend that the authors move this clarification into the main text so readers can immediately understand what components are being optimized and how gradients flow.

- **On the mathematical clarity and internal consistency**. While the paper’s theoretical framing is interesting, I found the mathematical presentation scattered and internally inconsistent. Symbols are introduced but never used or formally defined. For example, T(x) is defined once (Eq. 5) and never referenced again. The trace score s(x), which seems to be the central quantity guiding the policy updates, is described conceptually but never expressed mathematically. In addition, a reward r(x) is defined (Eq. 6); if I understood correctly, it corresponds to −s(x), but this relationship is never made explicit. In Eq. (7), it is also unclear what pϕ denotes—whether it is simply the policy distribution πϕ or a learned variant of the prior p0. The term Dpub is briefly defined as “a publicly available dataset with a similar distribution,” but its operational role remains vague. Is Dpub the same dataset used to pretrain the DCGAN generator, or is it only used for the KL regularization term? Clarifying this would help connect the regularized RL objective to the actual implementation.

- **on distribution-level comparison with baselines**. While the paper reports FID and KL metrics for ReTrace across datasets and access levels, it does not provide corresponding values for baseline methods (e.g., UIA, HRec). Since FID and KL are the primary metrics used to evaluate distribution-level reconstruction quality, the lack of direct comparison makes it difficult to assess whether ReTrace actually improves over existing approaches in recovering the overall deleted-data distribution.

**Questions:**

I would appreciate it if you could also answer my questions:

**Q1**. I might have missed it in the paper, but I don’t fully understand— as also mentioned in my weakness section—whether the optimization in Equation (7) aims to maximize the trace score 𝑠(𝑥) or the reward 𝑟(𝑥) and what the motivation is for defining both and how they are related.

**Q2**. My understanding is that, according to Equation (9), the instance-level reconstruction step selects a single top-scoring candidate via arg max 𝑠(𝑥) and refines that sample. If that is correct, could the authors clarify how the multiple instances shown in Figure 4(a) are reconstructed? A related question: did the refinement step lead to a significant improvement in reconstruction quality?

**Q3**. I noticed that DCGAN is used as the generative model throughout the experiments. Could the authors elaborate on the reasoning behind this choice? Given the many stronger generative models introduced in recent years (e.g., StyleGAN, diffusion models), wouldn’t using a more advanced generator potentially improve reconstruction quality?

**Q4**. In the appendix, it’s mentioned that the DCGAN produces 32 × 32 images. If I understood correctly, some of your evaluation datasets (e.g., Food-101) are higher-resolution. Could you clarify how this resolution mismatch is handled? In particular, how are comparisons with baselines such as UIA and HRec made—do these methods also operate at 32 × 32 resolution, or were their outputs downsampled to ensure fairness?

As I mentioned before, I like the core idea and find it creative and promising. However, I would need the authors to clarify the points raised in the weakness section—particularly by adding distribution-level comparisons against baselines—and address the questions above before I could confidently give this paper a clear accept.

---

> ### Author Response · Authors · 2025-11-23
> **Response to Reviewer MuvD (1/3)**
>
> We thank the reviewer for admitting our novel idea and timely contribution. We are delighted that the reviewer enjoyed reading our paper. We are confident to address all the raised weaknesses and answer the questions.
>
> We first address the concerns in weakness.
>
> **W1. Clarification of RL formulation**.
> We thank the reviewer for carefully pointing out the ambiguity in the RL formulation. To clarify, the RL agent’s action space is z, which is then passed through the pretrained generator $G$ to produce samples $x=G(z)$. The reviewer’s understanding and interpretation of ReTrace is completely correct. We fully agree that the architectural detail currently described in Appendix D.1 is essential for understanding the RL loop, gradient flow, and optimization mechanics. In the revised version, we will move this clarification into Section 3.2 (method description) so that readers can immediately see how states, actions, and policies are defined and how they interact with the generator. This adjustment will make the RL formulation more explicit and eliminate potential ambiguity.
>
> **W2. Mathematical clarity**.
> We really appreciate that the reviewer dive into our mathematical formulations and equations. We address the concerns one by one regarding this point.
> - Explanation of $T(x)$, $r(x)$, and $s(x)$.
> In our framework, $T(x)$ denotes the raw unlearning trace vector extracted from $f^+$ and $f^-$, whose dimensionality depends on the access level. Each component of $T(x)$ measures a different discrepancy between the  $f^+$ and $f^-$ (prediction shift, loss gap, or gradient deviation). For instance, in white-box setting, it contains all three access level traces. $T(x)$ serves as a bridge to help understand how Eq.6, i.e., $r(x)$, is obtained, so it does not show up in the later equations.
> $r(x)$  is the raw trace signal produced by our scoring functions (e.g., prediction discrepancy, loss difference, or gradient deviation). Because these quantities are unbounded and may vary significantly across samples, $r(x)$ may take values on very different numerical scales depending on the dataset, model architecture, and access level. PPO, however, requires rewards that are numerically stable, as overly large or highly variable rewards are known to cause gradient explosion, unstable policy updates, and slower convergence. Therefore, directly optimizing over $r(x)$ is inappropriate for our RL–based approach.
> To address this, we apply a simple monotonic min–max normalization over the candidate batch: $s(x) = \frac{r(x) - \min\_{x' \in \Pi} r(x')}{\max\_{x' \in \Pi} r(x') - \min\_{x' \in \Pi} r(x')}$, where $\Pi$ denotes the set of sampled candidates in the current PPO iteration. This transformation preserves ordering and ensuring $s(x) \in [0,1]$. Figure 1 visualizes this mapping, illustrating that PPO optimizes over the normalized trace score $s(x)$ by mapping $r(x)$. We will add this clarification in our revision to eliminate any ambiguity.
> - Clarification of Eq.7. We thank the reviewer for requesting our clarification of the terms in Eq.7. We explain the key terms in detail. There is a minor typo in Eq.7 that the $r(x)$ should be $s(x)$, as $s(x)$ is the final trace score used as reward,  but this does not affect any derivations, proofs, or experimental results in our paper. We clarify that $\pi\_\phi$ and $p\_\phi$ refer to the same object, i.e., the generator policy parameterized by $\phi$. $\pi\_\phi$ denotes the policy mapping from latent space to candidate samples, while $p\_\phi$ denotes the induced sample distribution over generated data. We will unify notation in the revision to avoid confusion. $\mathcal{D}\_{\text{pub}}$ is the same dataset used to pretrain the DCGAN generator in our reconstruction attack (e.g., under approximate unlearning scenarios). Thus, the KL regularization term $D(p\_{\phi}\,\|\,\mathcal{D}\_{\text{pub}})$ explicitly encourages the generator to stay close to the public data distribution.
> We also highly encourage the reviewer to see our response to reviewer FHN1 regarding weakness 3 (W3) and question 5 (Q5) for more insight and understanding.

---

> ### Author Response · Authors · 2025-11-23
> **Response to Reviewer MuvD (2/3)**
>
> **W3. Distribution-level comparison with baselines**.
> We thank the reviewer for suggesting that reporting FID and KL for baseline methods. To the best of our knowledge, ReTrace is for the first time to enable distribution-level reconstruction attacks in machine unlearning. Since UIA and HRec can only perform instance-level attacks under the white-box setting, we do not include the comparison of them and ReTrace on distribution-level reconstruction. However, we also agree with the reviewer that to make the results more convincing, providing the comparison on distribution-level reconstruction is valuable, even though the baseline methods are not designed for distribution-level attacks. We have already conducted additional experiments, and the results show that ReTrace significantly outperforms both two baselines. We provide a table below to show the results. Also, we will include the detailed results and analysis in our revised paper.
> | Dataset   | Method | FID (Exact)      | KL (Exact)            | FID  (Approximate)        | KL  (Approximate)               |
> |-----------|--------|----------------------------------|-----------------------------------|-----------------------------------|----------------------------------|
> | CIFAR-100 | UIA    | 203.6      | 5.2          | 189.7         | 4.8             |
> | CIFAR-100 | HRec   | 230.4     | 5.3           | 201.4         | 4.9              |
> | Food-101  | UIA    | 224.8    | 5.6           | 212.3         | 5.1              |
> | Food-101  | HRec   | 241.6    | 5.3           | 226.6         | 5.1              |
> | PathMNIST | UIA    | 156.4     | 4.1           | 143.4         | 3.8              |
> | PathMNIST | HRec   | 148.2    | 4.3     | 139.4         | 4.0              |
>
> We then answer the questions.
>
> **Q1. Optimization in Eq.7**.
> Eq.7 aims to maximize the trace score $s(x)$, which serves as the reward. We have clarified this by explaining the relationship between $s(x)$ and $r(x)$. We refer the reviewer to our response to W2.
>
> **Q2. Explanation of the refinement step in Eq.10**.
> We thank the reviewer for pointing out the refinement step in our approach. The reviewer is correct based on the understanding of Eq.9. To make it more clear, we explain the reason we use refinement. Eq.9 computes the trace score for all candidate samples $x\_i$ and selects only the highest-scoring one as the initial reconstruction $\hat{x}\_0$. Since this selection is based on a single sample, it may occasionally yield noisy, imprecise, or visually unnatural outputs, for example, an image whose color distribution resembles the forgotten instance but does not belong to the same class, or one that partially contains features of the forgotten class but is dominated by another category. This motivates the refinement step in Eq.10, which aims to further adjust $\hat{x}\_0$ so that the reconstructed sample becomes both more faithful to the forgotten content and more natural. Therefore, the refinement is significant to improve the reconstruction quality.

---

> ### Author Response · Authors · 2025-11-23
> **Response to Reviewer MuvD (3/3)**
>
> **Q3. Reasoning behind using DCGAN as generative model**.
> Thanks for the interesting question. We selected DCGAN for the following three reasons.
> - Our target is to evaluate the efficacy of ReTrace, so the generative model is just a final step to generate forgotten images. DCGAN serves as a widely adopted and well-understood generative model, allowing us to isolate and validate the effectiveness of the trace-guided reconstruction mechanism itself.
> - DCGAN offers an ideal balance between reconstruction fidelity and computational efficiency. It trains quickly, is stable across datasets, and enables extensive experimental sweeps over access levels, datasets, and unlearning methods. This practicality is essential for a systematic evaluation of ReTrace. While training StyleGAN or diffusion models is much more expensive.
> - Although architectures such as StyleGAN or diffusion models may yield more visually appealing samples, they only affect image realism, not the existence or exploitability of unlearning traces. ReTrace operates before the generator, on the discrepancy signals between unlearned and retained model, and would therefore benefit similarly from a stronger generator. Thus, replacing DCGAN would likely improve visual quality but would not alter the key finding: forgotten samples can be reconstructed due to residual traces.
> We will add a discussion section in our revised paper to discuss this point.
>
> **Q4. Clarification of image resolution used in experiments**.
> We thank the reviewer for this thorough question.
> - All evaluation datasets including higher-resolution ones such as Food-101 (224×224) and PathMNIST (28×28) are first preprocessed by uniformly resizing images to 32×32. This ensures that the input resolution matches the output resolution of the pretrained DCGAN generator and keeps the reconstruction task well-posed. Such resolution alignment is a standard practice in generative-model evaluation and does not alter experimental validity.
> - Regarding comparisons with baselines, UIA and HRec natively produce 32×32 reconstructions, so no additional downsampling or upsampling is needed. Therefore, all methods are evaluated under identical resolution, ensuring a fair and controlled comparison.

---

### Official Review · Reviewer_gE9i · 2025-11-02

**Soundness:** 3
**Presentation:** 4
**Contribution:** 3
**Rating:** 6
**Confidence:** 2

**Summary:**

The paper presents an attack on machine unlearning using RL-based data reconstruction. The work uses residual learnings as rewards and is comprehensively evaluated on several data samples using blackbox access on standard benchmark datasets. The findings indicate the feasibility of performing such attacks on large scale datasets.

**Strengths:**

- important problem and timely issue
- well-carried out attack framework without much prior assumptions
- good comparison on several benchmarks and good theoretical foundation

**Weaknesses:**

- comparison is limited to a few models and datasets and can easily be expanded broader
-  Convolution Transpose,instance-wise unlearning,  or Masked Small Gradients methods could have also been studied
- would have been great to mention computational costs and complexities

**Questions:**

The paper presents a good study and evaluation of hidden traces and connections leading to success in attacks against unlearning models. The work is well presented, though some of the popular models and approaches for unlearning have not been explored. I wonder if the comparison can benefit from the larger set of models and methods studied in "Deep Unlearn: Benchmarking Machine Unlearning for Image Classification" in EuroS&P'25.

What defenses can be used to mitigate the attacks mentioned in the paper?

It would be great to discuss the complexities of the attack and if it will be realistic to carry out against large datasets and bigger models.

---

> ### Author Response · Authors · 2025-11-23
> **Response to Reviewer gE9i (1/2)**
>
> We thank the reviewer for admitting our effort and providing the constructive and insightful comments. We are confident to address all the concerns and raised questions. We summarize them and answer each of them one by one.
>
> **W1. Generalization and scalability**
> Our work focuses on revealing the hidden trace signals that persist across unlearning pipelines and demonstrating how these signals can be exploited by reconstruction attacks. As a result, our experiments were intentionally scoped to representative machine unlearning settings (exact retraining and approximate SGU), which are the most widely adopted forms in prior studies [1, 2].
> Nonetheless, we are happy to compare our findings with a larger set of unlearning mechanisms, which could further strengthen the empirical evidence.
> We have expanded our experimental evaluation in the revised version. Specifically, we will incorporate (1) an additional architecture (ResNet-34), and (2) three widely used approximate MU methods including, Influence Unlearning (sparsity-based) [3], Boundary Unlearning [4], and Convolution-Transpose Unlearning [3]. These additional experiments will allow us to validate that the trace-based reconstruction signal exploited by ReTrace persists across a broader spectrum of architectures and unlearning implementations. The results are listed below, and will be included in our revised paper.
>
> | Method                          | Black-box MSE  | Black-box SR  | Black-box CS  | Grey-box MSE  | Grey-box SR  | Grey-box CS  | White-box MSE  | White-box SR  | White-box CS  |
> |---------------------------------|-----------------|----------------|----------------|----------------|----------------|----------------|------------------|-----------------|----------------|
> | Sparsity-based Unlearning       | 0.21          | 53.1%          | 0.46           | 0.20         | 63.8%          | 0.47           | 0.18           | 70.9%           | 0.49           |
> | Boundary Unlearning             | 0.20          | 53.5%          | 0.47          | 0.19         | 65.2%          | 0.49          | 0.17           | 71.3%           | 0.51           |
> | Convolution-Transpose Unlearning| 0.22          | 53.4%          | 0.46          | 0.21         | 63.2%          | 0.49          | 0.18           | 73.4%           | 0.50          |
>
> Result on Resnet-34:
> | Model     | Black-box MSE  | Black-box SR  | Black-box CS  | Grey-box MSE  | Grey-box SR  | Grey-box CS | White-box MSE | White-box SR | White-box CS  |
> |-----------|----------------|----------------|----------------|----------------|---------------|----------------|-----------------|----------------|----------------|
> |ResNet-34 | 0.21           | 60.2%          | 0.48           | 0.19           | 68.3%         | 0.50           | 0.15            | 74.9%          | 0.52           |
>
>
> **W2. Potential defense mechanism**
> We thank the reviewer for raising this concern. Unlearning inevitably leaves systematic traces, which is exploited exactly by ReTrace. Therefore, many existing traditional defenses that attempt to hide gradients, limit access, or perturb the output distribution are insufficient, because they do not eliminate the underlying model drift that unlearning necessarily induces. Specifically, Traditional defenses do not meaningfully alter the discrepancy $f^+(x)$ and $f^-(x)$. Even with heavy output noise, our grey-box and black-box results show that the trace signal remains detectable as long as the unlearning procedure changes the model in a structured manner. This aligns with our theoretical analysis: any method that truly removes forgotten data must shift either the loss landscape, the representation space, or the gradient behavior, and such shifts are precisely what ReTrace captures.
> Our discussion on potential limitations (Section Appendix D.6) indicates that weakening the traces can make the attack harder. As a result,  defenses may reduce attack success if they can reduce structural dependence between $f^+$ and $f^-$. For example, by re-initializing certain layers before retraining or using partial-model re-training to weaken the discrepancy signal. This results in the reward in RL becoming weaker in ReTrace, and thus makes the reconstruction not that effective.
> Overall, our results highlight that current unlearning mechanisms still induce detectable and exploitable traces, even when architectures, datasets, and access levels differ. ReTrace opens a path for future work toward more robust unlearning mechanisms and principled defense strategies.

---

> ### Author Response · Authors · 2025-11-23
> **Response to Reviewer gE9i (2/2)**
>
> **W3. Computational cost and complexities analysis**
> In Section 3.4, we have already provided a computation cost discussion theoretically, showing that the overall runtime of ReTrace is $O(I \cdot N \cdot C\_f + N \log N)$ dominated by generator sampling and trace evaluation across RL iterations with $N$ candidates per iteration. In Appendix B, we further explain the reason for this computation complexity from the perspective of our approach design.
>
> To make the computational cost more intuitive, we provide a table below, showing that both PPO optimization and DCGAN generation introduce only minimal computational overhead (≤0.02 s per step), confirming the practical efficiency of Retrace.
>
> | Step                                 | Average Time |
> |-------------------------------------|--------------|
> | PPO training (per optimization step) | 0.02s |
> | DCGAN each generation  | 0.005s |
>
> **Reference**
>
> [1] Hu, H., Wang S., Dong T., & Xue, M. Learn what you want to unlearn: Unlearning inversion attacks against machine unlearning. In 2024 IEEE Symposium on Security and Privacy (SP), pp. 3257–3275. IEEE, 2024.
>
> [2] Thudi, A., Deza, G., Chandrasekaran, V., & Papernot, N. Unrolling sgd: Understanding factors influencing machine unlearning. In 2022 IEEE 7th European Symposium on Security and Privacy (EuroS&P) (pp. 303-319). IEEE.
>
> [3] Cadet, X. F., Borovykh, A., Malekzadeh, M., Ahmadi-Abhari, S., & Haddadi, H. (2025, June). Deep Unlearn: Benchmarking Machine Unlearning for Image Classification. In 2025 IEEE 10th European Symposium on Security and Privacy (EuroS&P) (pp. 939-962). IEEE.
>
> [4] Chen, M., Gao, W., Liu, G., Peng, K., & Wang, C. (2023). Boundary unlearning: Rapid forgetting of deep networks via shifting the decision boundary. In Proceedings of the IEEE/CVF Conference on Computer Vision and Pattern Recognition (pp. 7766-7775).

---

### Official Review · Reviewer_FHN1 · 2025-11-04

**Soundness:** 2
**Presentation:** 2
**Contribution:** 2
**Rating:** 4
**Confidence:** 4

**Summary:**

The paper proposes RETRACE, a reconstruction attack on machine unlearning that frames recovery of deleted data as a reinforcement-learning (RL) problem. The method extracts “unlearning traces” by contrasting a pre-unlearning model  with a post-unlearning model at prediction, loss, and gradient levels, and uses these as rewards to train a generator. The theory claims the RL objective converges to an exponential-tilted policy that emphasizes high-trace regions; experiments report strong instance-level recovery and improved distributional alignment (lower FID/KL) versus UIA and HRec, plus preliminary text results with DistilBERT.

**Strengths:**

1. The paper proposes a new solution based on RL to address a newly proposed threat model against machine unlearning.
2. The paper considered both instance-level machine unlearning and class-unlearning setting in their approach.
3. Authors consider three levels of access to the original model and retrained model (black-box, grey-box, and whitebox), and perform most their evaluation on the three settings.

**Weaknesses:**

1. Some notations are used without proper definitions and clarifications. For example in line 141, $x_{\pi_{(j)}} is not defined.
2. Some of the details of the proposed approach are missing, which makes it confusing for the reader. What is the initial $z$ that you use in PPO? The details are missing while I think it might affect the outcome. I think the paper would benefit more from a pseudocode instead of figure 1 that does not provide any useful information.
3. From assumptions 1 and 2 of the paper, it seems authors assume access to the population data of the forget class, which is a more restrictive assumption of access to the population data of the original training set as in prior work [1].
4. In general, while this paper builds on an earlier paper for its setting [1], I don’t believe the setting of the attack specifically is specifically relevant to unlearning, but it is more in general about getting two models, one of which is trained on a subset of the training data used for the other, and try to reconstruct that data. So I think it is basically a privacy attack using model differentiation.  That is also why in [1], they only use the retrained model for their evaluations. I believe the assumption about gaining access to the original model in the setting of unlearning is not realistic.
5. Although the link for the code has been added and reviewers are referred to in the reproducibility statement, the repository was empty and only the score computations method (equation 5) was provided.
6. Your generative model relies on a DCGAN for generating the data that is missing. However, the pretrained DCGAN data is already trained on the unlearned classes and is capable of generating samples from the unlearned class. Therefore, it wouldn’t be surprising if DCGAN is capable of generating samples whose MSE from the original forget sample fall within the same range as the variance of the samples of that class. Basically, if I want to rephrase what you point out about the MSE value it would be sth like: you have a sample from the ‘Bed’ class that has been removed from the training set. Then you use your method to generate the image of a sample that can be considered a ‘Bed’, but not necessarily the same sample that was unlearned. Given that DCGAN is able to generate images of ‘Bed’ achieving that would not be very surprising. I think some of the confusion about this could be resolved once you respond to weakness 2, specially if based on assumption 1 of the paper, the adversary starts with a prior for the unlearned class.
7. I think the provided figures are not very informative as they are now. For example, looking at figure 8, I don’t see any significance on the values of the selected squares. Even in figure 2 the advantage of white-box over black-box is not clear from the figure. In white box all the values seem to be larger, not only the forgotten samples.
8. The results in the table are not accompanied by standard deviations. For example in table 1 the difference in CS score or MSE score for white-box vs grey-box is only at most 0.02, which might simply be smaller than the variance of the data.
9. Your theoretical results rely on the assumption that the expectation of the score assigned to the samples from the forgotten samples are strictly larger than this expectation for the retained data. However, the score that you defined in equations 2,3, and 4 do not seem to necessarily follow this assumption. For example, if you train a model on the retained data and train a model on the whole data, the loss of these models, should be more similar on the retrained data compared to the forget samples. I think this assumption should be at least accompanied by some empirical observations.

[1] Bertran, M., Tang, S., Kearns, M., Morgenstern, J. H., Roth, A., & Wu, S. Z. (2024). Reconstruction attacks on machine unlearning: Simple models are vulnerable. Advances in Neural Information Processing Systems, 37, 104995-105016.

**Questions:**

1. In the setting of the problem, it is assumed that the adversary has access to the auxiliary public dataset $D_{\mathrm{pub}}$. However, it is not clear from the paper what the initial $z$ in the experiment is? It has to be clarified what initial $z$ is chosen in the optimization (the initial state). Could the authors please elaborate on the details (with specific examples on the dataset and forgotten sample ideally)?
2. Have you tested your method on a class that DCGAN has not been trained on? What would happen in that case?
3. Could the others provide some plots on the number of iterations used in PPO and how the metrics they use (e.g., MSE) changes along this optimization?
4. Could the authors provide normalized values in figure 2 to show-case the improvement due to more information in the white-box setting. I would suggest computing the average value for all the 25 images in the patch and then reporting the ratio of the values for the red squares over the computed average value for the corresponding patch.
5. To maximize equation 7, as mentioned in line 214, you need to maximize the reward given in equation 6. For that you need to minimize the differences (due to negative signs). But this would mean samples that the retrained model and original model would act similar on (which would be the retained data) and for example for the retrained data, we would expect equation 2,3, or 4 achieve the smallest values. So why the model should converge to the forget samples. Could the authors please address this confusion I had when reading the approach?
6. In line 836, you mention exact unlearning method is implemented by “fine-tuning the model on the remaining data for the same number of epochs as the original training”. Why not training the model from scratch instead of fine-tuning the model on the remaining data. In practice the exact unlearning models are derived by fine-tuning the model from scratch because the fine-tuned model still contains information about the forget data and is not equivalent to the retrained model.
7. Currently the authors only evaluate the effectiveness of their method that rely on SGD update (either gradient descent on remaining samples or gradient ascent on the forget samples). It would be interesting to see the effectiveness of the attack on the two following settings that are shown to be more successful than GA:
    - Using sparsification methods that only perform SGD updates on a subset of the parameters [1,2].
    - Unlearning methods that do not rely on SGD on either of the remaining sets and forget sets [3,4].

[1] Jia, J., Liu, J., Ram, P., Yao, Y., Liu, G., Liu, Y., ... & Liu, S. (2023). Model sparsity can simplify machine unlearning. Advances in Neural Information Processing Systems, 36, 51584-51605.
[2] Fan, C., Liu, J., Zhang, Y., Wong, E., Wei, D., & Liu, S. (2023, October). SalUn: Empowering Machine Unlearning via Gradient-based Weight Saliency in Both Image Classification and Generation. In The Twelfth International Conference on Learning Representations.
[3] Chen, M., Gao, W., Liu, G., Peng, K., & Wang, C. (2023). Boundary unlearning: Rapid forgetting of deep networks via shifting the decision boundary. In Proceedings of the IEEE/CVF Conference on Computer Vision and Pattern Recognition (pp. 7766-7775).
[4] Ebrahimpour-Boroojeny, A., Sundaram, H., & Chandrasekaran, V. Not All Wrong is Bad: Using Adversarial Examples for Unlearning. In Forty-second International Conference on Machine Learning.

---

> ### Author Response · Authors · 2025-11-23
> **Response to Reviewer FHN1 (1/6)**
>
> We thank the reviewer for the constructive and insightful comments. We are confident that we can address the concerns raised in the weaknesses section and are willing to revise the paper accordingly. Below, we first provide point-by-point clarifications and solutions to the issues in the weaknesses, followed by detailed responses to the reviewer’s additional questions.
>
> **Responses to weaknesses**
>
> **W1. Notation clarification**.
> We thank the reviewer for pointing out the missing definition of the notation x_{\pi(j)}. We have now clarified the notation used in line 141 to make the MMD fully precise.
> Let $D_{\text{del}} = \lbrace x_1, \ldots, x_k\rbrace$ be the deleted set with an arbitrary but fixed indexing.
> Let $\Pi$ denote the set of all permutations over $\lbrace1, \ldots, k\rbrace$.
> For a reconstructed set $\hat{X} = \lbrace\hat{x}\_1, \ldots, \hat{x}\_k\rbrace$, the minimum matching distance is defined as
> $ \mathrm{MMDist}(\hat{X}, D\_{\text{del}}) = \min\_{\pi \in \Pi} \frac{1}{k} \sum\_{j=1}^{k}
>     d\_{\mathcal{X}}(\hat{x}\_j, x\_{\pi(j)})$, where $x\_{\pi(j)}$ denotes the element in $D\_{\text{del}}$ matched to $\hat{x}\_j$ under permutation $\pi$.
> This clarification does not affect any theorem statement or proof, and we will include the updated definition in the revised version.
>
> **W2. Initial $z$ used in PPO**.
> Our PPO agent operates in the latent space of the generator, meaning that PPO is used to optimize the latent code $z$.
> At the beginning of the optimization, the latent vector is sampled from a standard normal distribution: $z^{(0)} \sim \mathcal{N}(0, I)$. PPO effectively performs optimization over the generator’s latent space. To verify that initialization does not influence the outcome, we conducted a robustness study across 5 random seeds. The variance of the final reconstruction MSE remains low, demonstrating that RETRACE is initialization-invariant. The results are listed below.
>
> Figure 1 illustrates the overall workflow of ReTrace, giving a general understanding of each step. To understand our approach in more detail, we will also include a full algorithmic pseudocode for the reconstruction process in our revised paper, covering initial latent sampling, reward computation for each step, PPO update loop, and generation of the reconstructed sample, making the pipeline clearer and much easier to follow.
>
> | Metric | Value |
> |--------|-------|
> | Mean reconstruction MSE  | **0.2338** |
> | Standard deviation MSE | **0.0201** |
>
>
>
> **W3. Assumption of adversary’s public dataset access**
> We appreciate the reviewer’s attention to our assumptions. We clarify this concern from three aspects.
> - The adversary cannot access the unlearned or original dataset used for training $f^+$, so they are not capable of knowing the exact classes, let alone the population. What the adversary realizes is only the task domain (e.g. food classification), and then they can access the real-world public datasets containing numerous foods.
> - The phrase “an auxiliary public dataset $D_{pub}$ drawn from the same distribution” in the threat model is only intended to indicate that the auxiliary public data can belong to the same semantic class as the forgotten sample (e.g., public “apple-like” images for the apple class), which does not imply population-level access to the class-conditional distribution.
> Additionally, Assumption 1 does not state that the adversary has access to the population distribution of the forgotten class, either. It only constrains the function class of the policy $\pi$ and is standard in KL-regularized RL theory. It requires that $\pi$ is rich enough to represent certain distributions, not that the adversary can observe or sample from them.
> - Since in our threat model, $D_{pub}$ is a finite public dataset, not oracle access to the full data-generating distribution P(X|Y), this is fundamentally different from the assumption in Bertran et al. [1], who explicitly rely on population distributions and access to both pre-deletion and retrained models. The adversary in our scenario uses only class-consistent public samples, making our assumption substantially weaker and more realistic in unlearning threat models. We will clarify this wording in the revision to avoid potential ambiguity.

---

> ### Author Response · Authors · 2025-11-23
> **Response to Reviewer FHN1 (2/6)**
>
> **W4. Assumption of adversary’s access to original model**.
> We first clarify that access to the pre-unlearning model $f^+$ is not an unrealistic assumption, but a commonly occurring situation in the real world. As discussed in threat model (Section 2.2) and Appendix A, modern AI services frequently expose multiple model versions simultaneously due to
> - API versioning (e.g., OpenAI, Google Cloud Vision, Azure Cognitive Services);
> - Shadow and canary rollouts used by large-scale platforms (e.g. Meta and Google);
> - Rollback requirements in regulated domains such as finance and healthcare, where older checkpoints are preserved.
>
> These practices directly allow adversaries to observe both  $f^+$ and  $f^-$, making our threat model realistic and motivated by real production environments rather than theoretical convenience.
> Then we explain that our work is specifically relevant to unlearning. Our attack is intrinsically tied to the unlearning process, rather than a generic model differentiation attack. In our formulation, $f^-$ is not trained on an arbitrary subset of the data. Instead, it is obtained specifically through an unlearning mechanism, especially for approximate unlearning, and our reward design, trace signals, and empirical analyses all rely on discrepancies caused by the unlearning operation itself. Additionally, the work [1] titled explicitly “Reconstruction Attacks on Machine Unlearning”, is itself framed as an unlearning-specific reconstruction attack. The authors clearly position their setting as an attack on unlearning, not as a generic model-differentiation problem according to the title.
> For these reasons, our threat model is realistic, consistent with prior unlearning literature, and fundamentally different from generic model differentiation.
>
> **W5. Provided link for repository**.
> We have double checked the link provided in our submission, and we clarify that the provided repository is not empty. The implementation has been available since the initial submission at: https://anonymous.4open.science/r/ReTrace-FE5F.
>
> The repository includes:
> - README.md: a usage guide with concrete commands for training $f^+$ and $f^-$, and running the RL–GAN reconstruction;
> - RL_GAN.py: implementation of the RL–GAN reconstruction procedure used in our attack;
> - trace_distribution.py: code to compute and analyze the trace distributions, including the score in Eq.5;
> - trace_heatmap.py: code to extract unlearning traces and generate the heatmaps shown in the figures;
> - unlearning.py: scripts for training $f^+$, performing exact and approximate unlearning for $f^-$, and computing trace signals.
>
> **W6. Discussion of the pretrained DCGAN**.
> - Our use of a pretrained DCGAN that has seen the forgotten class is not an artifact but a direct instantiation of our threat model. The adversary knows the task domain and can access an auxiliary public dataset drawn from the same semantic distribution family. Therefore, it is realistic and reasonable for the pretrained prior (Assumption 1) to include samples from the forgotten class.
> - We have discussed the influence of using a pretrained model without an unlearned class in Appendix. As discussed in Appendix D.6, the alignment between the pretrained corpus and the forgotten classes affects reconstruction quality. If the GAN’s pretraining data diverges from the target domain, reconstruction quality may decrease. However, because large-scale pretrained generative models are standard in practice, and most domains have abundant public data, this influence is mild and does not undermine the validity of our method or experiments.
> - To fully address the reviewer’s concern regarding this point, we have conducted experiments where the pretrained DCGAN has not seen the forgotten class. We exclude class 1 in CIFAR-100 in the pre-training. The reconstruction quality decreases a bit for both levels’ reconstruction, but the attack still extracts meaningful structural cues from the traces extracted from $f^+$ and $f^-$. We list the results (MSE, SR, and CS for instance-level reconstruction; while FID and KL for distribution-level reconstruction) below, and will include them in our evaluation (Section 5) in the revised paper.
>
> | Method                | Access Level | MSE  | SR   | CS   | FID | KL |
> |----------------------|--------------|-------|--------|-------|-------|------|
> | **Exact Unlearning** | Black-box    | 0.33  | 42.1%  | 0.32  | 203.4  | 5.3 |
> |                      | Grey-box     | 0.30  | 43.7%  | 0.36  | 199.2  | 4.9 |
> |                      | White-box    | 0.31  | 45.4%  | 0.38  | 183.8  | 4.7 |
> | **Approx. Unlearning** | Black-box  | 0.27  | 51.3%  | 0.42  | 182.1  | 4.2 |
> |                      | Grey-box     | 0.27  | 58.5%  | 0.43  | 169.7   | 4.0 |
> |                      | White-box    | 0.24  | 59.6%  | 0.45  | 158.9   | 3.7 |

---

> ### Author Response · Authors · 2025-11-23
> **Response to Reviewer FHN1 (3/6)**
>
> **W7. Figures regarding the traces of unlearned data (Figure 2, 6, and 8)**.
> To explain and clarify this point, we answer the following four questions.
> - **What’s the purpose of these figures?** These figures are intended to illustrate the contrast between the traces left by unlearned data and retained (non-unlearned) data across the black-box, grey-box, and white-box settings. Overall, we observe that in both the grey-box and white-box settings, the traces of retained data exhibit a more concentrated distribution, with smaller variance, compared to the black-box scenario. At the same time, the traces of unlearned data also show tighter concentration, lower variance, and higher stability in the grey-box and white-box settings than in the black-box setting. The white-box setting has the smallest variance. This behavior is consistent with our experimental findings: grey-box and white-box provide richer adversarial information, which leads to more stable trace signatures and correspondingly higher reconstruction success rates. Importantly, the goal of these figures is to highlight the differences in trace patterns, rather than the absolute numeric values.
> - **Providing normalized values to show the improvement in the white-box setting?** We thank the reviewer for the helpful suggestion. We agree that providing a normalized version of Figure 2 can make the contrast between the black-box, grey-box, and white-box settings more visually explicit. We have computed, for each 5×5 patch, the average trace value and then get the ratio between the highlighted red-square values and the corresponding patch average. We have already reproduced this visualization and confirmed that the normalized ratios further amplify the relative contrast: the white-box setting shows the largest ratio gap between unlearned and retained samples, followed by the grey-box setting, while the black-box setting exhibits noticeably weaker ratios. This matches the quantitative behavior observed in our evaluation and reinforces the phenomenon we aimed to illustrate. We will include this normalized version in the revision.
> - **Why white-box values appear uniformly larger?** The trace values in the white-box setting appear higher. This is expected because when internal information of $f^+$ (e.g., intermediate activations and gradients) becomes directly accessible in the white-box setting, the computations (Eq.4) use more detailed signals. These internal signals may naturally tend to produce larger numerical outputs, which leads to an overall upward shift in the trace values. Importantly, this shift in absolute values does not affect the distinguishability between unlearned and retained samples. The relative patterns remain clearly different, and this difference is even more pronounced, consistent with its higher reconstruction success rate.
> - **Why does Figure 8 seem not informative?** For Figure 8, we intentionally included a few unlearned samples that exhibit weaker visual saliency (e.g., the two examples in the bottom-right) in order to faithfully represent real-world scenarios where some forgotten instances behave as outliers or have inherently low contrast in certain scenarios. However, as shown in Figure 9, when aggregating over all samples, the average trace score for unlearned data is substantially higher than that of retained classes, yielding a clear and consistent distinction even in more complex medical-image domains. Thus, the overall conclusion remains robust. To eliminate any ambiguity, we will update Figure 8 by randomly selecting unlearned samples, consistent with Figure 2 and 6, making the visualization more convincing.

---

> ### Author Response · Authors · 2025-11-23
> **Response to Reviewer FHN1 (4/6)**
>
> **W8. The results in Table 1 lack of std**. Our evaluation already spans three datasets of different scales and domains, and across all of them, the white-box setting consistently achieves reliably stronger reconstruction performance. This consistency across heterogeneous tasks indicates that the observed improvements are not accidental fluctuations, but reflect the true behavior of the method under increased model access level. In our evaluation, we ran 5 independent experiments, so the results presented in Table 1 reflect the mean of the 5 runs. Thus, we can further provide the standard deviation across these runs to confirm that the observed gains are not due to randomness but remain statistically robust. The results confirm that the improvements in the white-box setting persist under multi-run evaluation. Therefore, the white-box advantage is statistically stable rather than due to fluctuations. The results are presented as follows and we will incorporate these multi-run results and the corresponding std into our revised version.
>
> *Exact Unlearning Result:*
> | Dataset   | Black-box MSE | Black-box SR            | Black-box CS            | Grey-box MSE | Grey-box SR            | Grey-box CS            | White-box MSE | White-box SR            | White-box CS            |
> |-----------|----------------|---------------|---------------|---------------|---------------|---------------|----------------|---------------|---------------|
> | CIFAR-100 | 0.23 ± 0.05    | 52.3% ± 0.4%  | 0.43 ± 0.04   | 0.22 ± 0.03    | 58.7% ± 0.5%  | 0.46 ± 0.03    | 0.20 ± 0.02    | 62.4% ± 0.6%  | 0.47 ± 0.02    |
> | Food-101  | 0.25 ± 0.04    | 48.9% ± 0.5%  | 0.38 ± 0.04    | 0.23 ± 0.03    | 54.7% ± 0.4%  | 0.42 ± 0.02    | 0.23 ± 0.02    | 50.4% ± 0.6%  | 0.44 ± 0.02    |
> | PathMNIST | 0.26 ± 0.04    | 49.8% ± 0.5%  | 0.26 ± 0.03    | 0.23 ± 0.02    | 50.6% ± 0.4%  | 0.27 ± 0.02    | 0.22 ± 0.01    | 52.8% ± 0.5%  | 0.31 ± 0.02    |
>
> *Approximate Unlearning Result:*
> | Dataset   | Black-box MSE | Black-box SR            | Black-box CS            | Grey-box MSE | Grey-box SR            | Grey-box CS            | White-box MSE | White-box SR            | White-box CS            |
> |-----------|----------------|---------------|---------------|---------------|---------------|---------------|----------------|---------------|---------------|
> | CIFAR-100 | 0.24 ± 0.04    | 55.9% ± 0.6%  | 0.46 ± 0.03    | 0.21 ± 0.03    | 68.7% ± 0.4%  | 0.49 ± 0.03    | 0.17 ± 0.02    | 73.1% ± 0.5%  | 0.50 ± 0.02    |
> | Food-101  | 0.24 ± 0.03    | 56.6% ± 0.5%  | 0.41 ± 0.03    | 0.18 ± 0.02    | 62.5% ± 0.5%  | 0.49 ± 0.02    | 0.16 ± 0.01    | 65.3% ± 0.4%  | 0.49 ± 0.01    |
> | PathMNIST | 0.24 ± 0.03    | 51.9% ± 0.5%  | 0.28 ± 0.02    | 0.20 ± 0.02    | 54.6% ± 0.5%  | 0.28 ± 0.02    | 0.19 ± 0.02    | 59.7% ± 0.5%  | 0.33 ± 0.01    |
>
>
> **W9. Reliability of Def.2 regarding trace scores**.
> Our assumption that the expected trace score for forgotten samples is larger than that for retained samples is consistent with our score definition (Eqs.2–4) and with the actual behavior of unlearning through our empirical evaluation.
> - Eq.2 measures the prediction difference between $f^+$ and $f^-$ on the same input.
> Since unlearning primarily changes the model’s behavior on forgotten samples, the gap is naturally larger for forgotten data and smaller for retained data. In Eq.3 and 4, the components are directly tied to the fundamental difference between forgotten and retained samples under unlearning. Eq.3 captures the shift in the model outputs (logits) between $f^+$ and $f^-$. Since forgotten samples are excluded from the training of $f^-$, the output discrepancy is naturally larger on forgotten samples. Eq.4 measures the difference in gradient responses between $f^+$ and $f^-$. Because unlearning alters the decision boundary specifically around the forgotten samples, the gradient deviation is amplified for unlearned data.
> - Our visualizations (Figure 2, 6, and 8) show that forgotten samples consistently show higher trace values in general. Their distributions exhibit smaller variance and clearer concentration, while retained samples spread across lower trace-value regions. These empirical results already presented the reliability of Def.2.

---

> ### Author Response · Authors · 2025-11-23
> **Response to Reviewer FHN1 (5/6)**
>
> **Responses to questions**
>
> **Q1. Initial state and details**.
> We answer this question in our response to weakness 2. We refer the reviewer to the first point of W2.
>
> **Q2. Test method on a class that DCGAN has not been trained on**.
> We refer the reviewer to our response to W6.
>
> **Q3. MSE changes along the number of iterations used in PPO**.
> We thank the reviewer for the suggestion. To address this point, we have generated the PPO optimization curves that track how the reconstruction objective evolves as PPO iterates. Specifically, we log the MSE between the reconstructed sample and its counterpart under $f^+$ every 10 iterations.
> Across all datasets and three model access levels, the optimization curves exhibit the following consistent behaviors:
> - White-box setting reaches its optimum the earliest, achieving the lowest MSE thanks to the richer supervision signals available from $f^+$.
> - Grey-box achieves a similarly stable improvement trajectory, converging smoothly toward its optimum under partial information.
> - Black-box also demonstrates consistent and reliable improvement, with the characteristic that it requires more PPO updates due to its more restricted feedback structure.
> These observations confirm that PPO is indeed optimizing the reconstruction objective effectively. We will include these results in our revision.
>
> **Q4. Normalized values in Figure 2 to show-case the improvement in the white-box setting**.
> We refer the reviewer to our response to W7.
>
> **Q5. The model should converge to the forgotten samples.**
> We thank the reviewer for pointing out this. We clarify that Eq.6 defines the raw discrepancy components, whereas Eq.7 is intended to use the trace score $s(x)$ (i.e., a monotonic min-max normalization mapping of the raw discrepancy) as the PPO reward. This is consistent with Figure 1 (Mapping $\rightarrow s(x) \in [0,1] \rightarrow$ Reward) and with the variational formulation in Appendix C (Eq.24), which clearly shows that the PPO objective is driven by $s(x)$, not by the raw negative discrepancy.
>
> Eq.7 with $r(x)$ instead of $s(x)$ is a minor notational typo on our side, which may cause confusion. However, this typo does not affect any derivations, proofs, or experimental results, as all computations in practice use $s(x)$ as the reward. We will correct the notation in the revised version to avoid ambiguity.
>
> Moreover, unlearning induces substantially larger drift between $f^+$ and $f^-$ on unlearned samples than on retained samples. Concretely, for a forgotten sample $x_f$, we have large prediction drift (Eq.2), large confidence/loss drift (Eq.3), and large gradient drift (Eq.4), while for a retained sample $x_r$, these quantities remain small.
> Since $s(x)$ is constructed from these components, this implies $s(x\_f) \gg s(x\_r)$. Because PPO maximizes the expected reward, the optimal policy naturally places most probability mass on inputs that maximize s(x) (i.e., on forgotten samples rather than retained ones). This is exactly what the softmax form expresses: samples with higher trace scores (forgotten data) receive exponentially higher priority under the policy.
>
> **Q6. Exact unlearning clarification**.
> We evaluate Retrace under two unlearning settings (exact and approximate). For exact unlearning, our implementation strictly follows the protocol (named retraining) used in Hu et al., IEEE S&P 2024 [6] and Bertran et al., NeurIPS 2024 [7], which are widely adopted in recent machine-unlearning work.
> Concretely, for each dataset, we split the training set into a public subset
> $D\_{0}$ (80%) and a private subset $D\_{u}$ (20%). We first train a model $M\_{0}$ on $D\_{0}$, which acts as the pre-trained backbone. Then, we fine-tune $M\_{0}$ on $D\_{u}$ for several epochs to obtain the final model $M$. This setting simulates a common real-world practice where developers start from a public pre-trained model and adapt it to their private data, a setup widely used in DNN training and explicitly adopted in multiple unlearning papers. It has also been shown to reduce unlearning error in exact unlearning baselines. We follow Hu et al.’s definition: removing a sample $x$ from $D\_{u}$, and obtaining an unlearned model $M\_{u}$ by fine-tuning $M\_{0}$ on $D\_{u} \lbrace x \rbrace$ for the same number of epochs as the original training. This protocol is considered a correct instantiation of the retraining-based baseline.
> To avoid ambiguity, we will explicitly describe this detailed procedure and clarify that our implementation is fully aligned with prior authoritative unlearning works in the revision.

---

> ### Author Response · Authors · 2025-11-23
> **Response to Reviewer FHN1 (6/6)**
>
> **Q7. Evaluate ReTrace under more unlearning settings**.
> We thank the reviewer for encouraging us to test ReTrace on (1) sparsification methods that only perform SGD updates on a subset of the parameters and (2) methods that do not rely on SGD. We clarify that ReTrace is not tied to the SGD update rule. In fact, it does not assume or require any specific unlearning mechanism, while its only requirement is that the unlearning procedure leaves detectable traces in the model.
> Most unlearning methods leave traces during the unlearning process, which is evidenced by existing work [5]. This fact makes ReTrace broadly applicable and successful across diverse unlearning paradigms.
> Following the comment, we extended our evaluation to another three representative unlearning baselines, Influence Unlearning (sparsity-based) [1], Boundary Unlearning [3], and Convolution-Transpose Unlearning [8]. Their results are listed as follows, showing the effectiveness of our method.
>
> | Method                          | Black-box MSE  | Black-box SR  | Black-box CS  | Grey-box MSE | Grey-box SR  | Grey-box CS  | White-box MSE  | White-box SR  | White-box CS  |
> |---------------------------------|-----------------|----------------|----------------|----------------|----------------|----------------|------------------|-----------------|----------------|
> | Sparsity-based Unlearning       | 0.21          | 53.1%          | 0.46           | 0.20         | 63.8%          | 0.47           | 0.18           | 70.9%           | 0.49           |
> | Boundary Unlearning             | 0.20          | 53.5%          | 0.47          | 0.19         | 65.2%          | 0.49          | 0.17           | 71.3%           | 0.51           |
> | Convolution-Transpose Unlearning| 0.22          | 53.4%          | 0.46          | 0.21         | 63.2%          | 0.49          | 0.18           | 73.4%           | 0.50          |
>
> **Reference**
>
> [1-4] are consistent with those in the reviewer's comment.
>
> [5] Chen, Y., Pal, S., Zhang, Y., Qu, Q., & Liu, S. Unlearning Isn't Invisible: Detecting Unlearning Traces in LLMs from Model Outputs. In ICML 2025 Workshop on Machine Unlearning for Generative AI.
>
> [6] Hu, H., Wang S., Dong T., & Xue, M. Learn what you want to unlearn: Unlearning inversion attacks against machine unlearning. In 2024 IEEE Symposium on Security and Privacy (SP), pp. 3257–3275. IEEE, 2024.
>
> [7] Bertran, M., Tang, S., Kearns, M., Morgenstern, JH., Roth, A., & Wu, SZ. Reconstruction attacks on machine unlearning: Simple models are vulnerable. Advances in Neural Information Processing Systems (NeurIPS), 37:104995–105016, 2024.
>
> [8] Cadet, X. F., Borovykh, A., Malekzadeh, M., Ahmadi-Abhari, S., & Haddadi, H. (2025, June). Deep Unlearn: Benchmarking Machine Unlearning for Image Classification. In 2025 IEEE 10th European Symposium on Security and Privacy (EuroS&P) (pp. 939-962). IEEE.

---

### Author Response · Authors · 2025-11-28

Dear Reviewers,

Thank you again for your thoughtful and constructive review.

All the required additional experiments have been done. The results are consistent with what we have shown in our paper and discussed in rebuttal. The content that might cause ambiguity has also been fixed. We re-write the approach section to explain more detail about ReTrace's method, and also refine our theoratical analysis section to make it clearer.

We are now revising the paper, and once finished we will submit a revised version including all the points we discussed in our rebuttal as well as the new experimental results. The revised parts will be highlighted in blue.

Best regards,

Authors of Paper #14618

---

> ### Author Response · Authors · 2025-12-04
> **Revised Paper Updated**
>
> We thank the reviewers for the thoughtful and constructive review again. We have submitted our revised version. The changes are highlighted in BLUE.
> - We added all the required experiments in the revised paper and provided detailed explanation and analysis.
> - We re-wrote the approach section to make it easier understanding and more align with our threat model.
> - We refine our theoratical analysis section to remove ambiguity.
> - We reploted some figures and tables to make the result clearer and more convincing.
> - We fixed few minor typos to make the paper more readable.
>
> Best regards,
>
> Authors of Paper #14618

---

### Meta-Review · Area_Chair_DB4h · 2026-01-13

**Summary:**

This paper proposes ReTrace, an RL-based reconstruction attack on machine unlearning using trace signals from pre- and post-unlearning models. The idea of using RL to search the generator’s latent space guided by unlearning traces is novel and interesting (MuvD, gE9i), and it provides a unified framework for black-, grey-, and white-box settings. Empirically, ReTrace outperforms SoTA baselines, such as UIA and HRec (MuvD, FHN1).

The rebuttal significantly strengthened the paper. The authors added standard deviations, more unlearning baselines (e.g., sparsity-based, boundary, convolution-transpose), and extra architectures (ResNet-34), and clarified several technical details. These additions make the empirical results more convincing and show that the attack is not limited to a single setting. The RL and reward definitions were initially unclear (FHN1, MuvD), and were clarified and corrected in the rebuttal.

However, some concerns remain. The threat model (access to both pre- and post-unlearning models and public data) is plausible in some real systems but should be more carefully scoped (FHN1).  There is also some ambiguity between recovering a specific instance and generating a similar class sample, especially given the use of a pretrained GAN (FHN1, MuvD).

Overall, despite remaining concerns about clarity and threat-model scope, the approach is technically interesting, and supported by substantially improved empirical evidence. The AC recommends acceptance of the paper.

**Reviewer Concerns:**

Addressed: FHN1, MuvD, gE9i
Not fully resolved: tMmg

**Reviewer Scores:**

FHN1: 4->6
MuvD: 6->6
gE9i: 6->6
tMmg: 4->4

---

### Decision · Program_Chairs · 2026-01-26

Accept (Poster)